# Identification of a binding site on soluble RANKL that can be targeted to inhibit soluble RANK-RANKL interactions and treat osteoporosis

Dane Huang [1,2,3], Chao Zhao [1,3], Ruyue Li [2], Bingyi Chen [1], Yuting Zhang[1], Zhejun Sun [1], Junkang Wei [1], Huihao Zhou [1] ✉, Qiong Gu [1] ✉ & Jun Xu [1] ✉

One of the major challenges for discovering protein-protein interaction inhibitors is identifying selective and druggable binding sites at the protein surface. Here, we report an approach to identify a small molecular binding site to selectively inhibit the interaction of soluble RANKL and RANK for designing anti-osteoporosis drugs without undesirable immunosuppressive effects. Through molecular dynamic simulations, we discovered a binding site that allows a small molecule to selectively interrupt soluble RANKL-RANK interaction and without interfering with the membrane RANKL-RANK interaction. We describe a highly potent inhibitor, S3-15, and demonstrate its specificity to inhibit the soluble RANKL-RANK interaction with in vitro and in vivo studies. S3-15 exhibits anti-osteoporotic effects without causing immunosuppression. Through in silico and in vitro experiments we further confirm the binding model of S3-15 and soluble RANKL. This work might inspire structure-based drug discovery for targeting protein-protein interactions.

Protein-protein interactions (PPIs) inhibitors gain increasing attentions[1,2] in pharmaceutical sciences. Because most proteins must accomplish their functions through PPIs. One of the challenges, however, is to identify a druggable binding site for a molecule to selectively inhibit the PPIs[3,4]. Compared to the classic binding site of drug targets like enzymes or kinases, the PPIs interface is large, flat, and lacks a suitable size site for the small molecules binding with. Therefore, the small molecular PPIs inhibitors own large molecular weight and consist with many hydrophobic functional groups to form more interactions with their targets. These usually lead to poor pharmacokinetic properties and off-target effects[1]. Tumor necrosis factor superfamily (TNFSF) is a typical example that has such challenge. TNFSF superfamily plays important roles in the rheumatoid arthritis (RA), Crohn's disease (CD), inflammatory bowel disease

(IBD), psoriasis, atherosclerosis, bone loss, and Alzheimer's disease (AD)[5–7].

Many TNFSF antibodies such as adalimumab, infliximab, and denosumab have been employed for treating rheumatoid, osteoporosis, and osteosarcoma. However, the severe immune-related side effects (such as increased risk of infection, autoimmune diseases, cancer, and reactivation of tuberculosis) caused by these pan-PPIs inhibitory biologics limit their clinic values[8–10]. TNFSF proteins own two forms, membrane-bound form (mTNFSF) and soluble form (sTNFSF) that is cleaved from mTNFSF[11,12]. The main reason for those side effects is mTNFSF expressed as type II transmembrane proteins (receptors) in immune cells, such as antigen presenting cells and T cells. mTNFSF proteins function as immune enhancers through reverse signaling pathways[11,12]. On the other hand, sTNFSF proteins are

[1]Research Center for Drug Discovery, School of Pharmaceutical Sciences, Sun Yat-Sen University, Guangzhou 510006, China. [2]Guangdong Provincial Key Laboratory of Research and Development in Traditional Chinese Medicine, Guangdong Provincial Second Hospital of Traditional Chinese Medicine (Guangdong Provincial Engineering Technology Research Institute of Traditional Chinese Medicine), Guangzhou 510095, China. [3]These authors contributed equally: Dane Huang, Chao Zhao. ✉e-mail: zhuihao@mail.sysu.edu.cn; guqiong@mail.sysu.edu.cn; junxu@biochemomes.com

over expressed in disease status. Hence, the therapeutical agents targeting sTNFSF PPIs must be selective to sTNFSF without interfering mTNFSF PPIs. Antibodies cannot distinguish mTNFSF and sTNFSF per se because they blanketly wrap up the same PPIs binding domain of mTNFSF or sTNFSF. Therefore, scientists including FDA agents suggest to discover small molecular TNFSF PPIs inhibitors[13]. Suramin is a small molecular inhibitor that inhibits TNF activity by disrupting the TNF trimer[14]. Another small molecular TNF PPIs inhibitor is SPD304, which binds to the interface of a TNF dimmer, disrupts the TNF trimer, and inhibits the cytokine function[15]. Unfortunately, selective small molecular PPIs inhibitors are not reported so far.

Receptor Activator of Nuclear Factor-κB Ligand (RANKL) belongs to TNFSF family. Therefore, RANKL also exists soluble and membrane forms. Normally, the functions of soluble RANKL (sRANKL) and membrane RANKL (mRANKL) are similar in osteoclast differentiation. However, over-expressed sRANKL can cause excessive bone resorption that induces osteoporosis[16]. The osteoporosis patients with therapy exhibited decreased level of sRANKL. Interfering RANKL–RANK interaction can inhibit osteoporosis[17] or bone metastasis[18]. Denosumab, a human monoclonal antibody, is the only approved therapeutic agent targeting RANKL for treating postmenopausal osteoporosis, giant cell tumor of bone[17,19,20] and bone cancer metastasis[21]. For the same reason, denosumab can also cause severe immune side effects due to its blanketly wrapping up the same PPI binding domain of sRANKL and mRANKL. The mRANKL is also expressed in the surface of lymphocytes, like T cell and B cell[22]. When T-cells with antigen, mRANKL becomes a receptor of RANK-RANKL reverse signaling pathway[11,12] and enhances immune functions such as, T-cell proliferation, T-cell–dendritic cell (DCs) interactions, DCs survival, thymus and lymph node development[23]. Blocking mRANKL-RANK reverse interaction results in osteopetrosis due to the lack of osteoclast; defective T-cell and B-cell differentiation, a failure of mammary gland lobuloalveolar development during pregnancy, decrease monocytes, DCs survival and their effective function[24–26]. This mRANKL can be cleaved by TNF-α converting enzyme (TACE) to form sRANKL[27]. Many researchers report that sRANKL from B cell and T cell promotes osteoclastogenesis, subsequently leads to osteoporosis[28] and cancer metastasis[27,29]. In addition, the serum level of sRANKL was shown to be a significant risk predictor of type 2 diabetes mellitus (T2DM) in a large prospective study[30]. Therefore, the selective sRANKL inhibitor may avoid immune side effects.

A small molecule that can selectively inhibit sRANKL-RANK interactions is a therapeutic solution for treating osteoporosis avoiding immune side-effects. Until now, small molecular sRANKL-selective PPIs inhibitor is not discovered yet. The major premise of discovering such small molecular inhibitors is to identify the binding site which is druggable and can discriminate sRANKL from mRANKL.

Here, we hypothesize that protein thermal fluctuation can lead us to discover this binding site. One traditional method of PPIs inhibitors discovery usually based on the PPIs interface from x-ray structures[1]. As known, the x-ray structure usually presents the most stable conformation. However, the protein conformation is not constant, but fluctuant[31]. During this thermal motivation process, it may form a druggable binding site. On the other hand, the mRANKL is rigidified by the cell membrane while sRANKL has no C-terminal extracellular connecting stalk domain and, is free from the cell membrane[32]. Hence, the fluctuation may also distinguish the sRANKL and mRANKL.

Therefore, we applied a molecular dynamic (MD) study to mimic thermal fluctuation and identified a selective binding site on sRANKL based on our hypothesis. This site was further proved to be druggable by biostudies. Then, a series of sRANKL-RANK small-molecular inhibitors was discovered based on this site, confirming the inhibitor's selectivity toward sRANKL and, validating the efficacy of anti-osteoporosis activity of the sRANKL without causing immunosuppression side effects in vivo and in vitro. This work provides a selective sRANKL small molecule inhibitor as a promising anti-osteoporosis agent without immune side effects. It also might inspire the structure-based drug discovery targeting PPIs for other drug target families.

## Results

### Identifying small molecule binding site that can be used to distinguish sRANKL from mRANKL

sRANKL is a non-covalently formed homotrimer[33], and cleaved from mRANKL. mRANKL has C-terminal extracellular connecting stalk binding to the membrane. Hence, a binding site of the homotrimer is induced easily for a small molecule[32] to bind sRANKL rather than mRANKL. Therefore, our initial effort attempts to identify selective binding sites for disturbing sRANKL-RANK interaction by molecular dynamics (MD) simulations. Firstly, A homology model of mouse mRANKL was created based on a mouse sRANKL crystal structure (PDB ID: 1S55)[34] by freezing the stalk amino acids close to the membrane because the crystal structure of mRANKL was not available. Then, 100 ns MD simulations on the models of sRANKL and mRANKL based on crystal a structure were conducted. The MD simulations results are documented in supplementary material (Supplementary Fig. 1).

In sRANKL or mRANKL, three key residues (K180, I246, and Q236 in mouse RANKL) are regarded as key residues for RANK binding[35]. Thus, we hypothesize that a druggable binding site that can be used to distinguish sRANKL from mRANKL should be close to these key residues. Therefore, we focus on the variation of Y234, Q236, K180, Q291, H252, I246, W192, Q302, D301, D299, R222, and H224 (Fig. 1a) during MD simulation. The MD results indicated four key residue-pairs (Y234-H224, Y234-D301, K180-H224, and Q236-D301) demonstrated different behaviors in sRANKL and mRANKL. The distances of these residue-pairs stand for the flexibility and size of the binding site.

To examine the flexibility of the binding site, the frequencies of the four residue-pair staying in long-distance status (> 2 Å) during the 100 ns MD simulations were studied in Fig. 1b, which indicated these residue-pairs significantly stayed at longer distance statuses in mouse sRANKL (red bars) rather than the ones in mouse mRANKL (blue bars).

To further examine the size of the binding site, we also defined the sum of the distances of the four residue-pairs (SDRP). Longer SDRP means higher flexibility and larger size and ligand-inducibility. The distance distribution of SDRP during the 100 ns MD simulations is depicted in Fig. 1c, which demonstrates that the SDRP in mRANKL (blue bars) tended staying shorter distances rather than the one in sRANKL (red bars). The common conformation of sRANKL (SDRP is 76–78 Å) and mRANKL (SDRP is 70–72 Å) were selected (Fig. 1d, e). The size of site was measured by distance of Y234-D303 and K180-R222 that present the length and width. The results revealed that the site size in sRANKL is much larger than mRANKL.

To experimentally confirm in silico results, the single-point mutation experiments on rat sRANKL[36] were conducted (The residue ID of rat RANKL is 2 more than that of mouse RANKL, like rat sRANKL K182 is equivalent to mouse sRANKL K180). We mutated K182A, R224A, H226A, E227A, Q238A, Y242A, E270, D301A, and D303A. Since these residues were possible to form the binding site and play important roles for binding with RANK[37,38]. Osteoclastogenic assays were conducted on these mutated and the wild type (WT) RANKL proteins. As shown in Fig. 1f–h, these mutated sRANKL proteins were not able to promote osteoclasts except Q238A. Thus, K182, R224, H226, E227, Y242, E270, D301, and D303 are experimentally confirmed to be important for RANKL-RANK binding and inhibiting osteoclastogenesis. In short, a binding site of RANKL for developing anti-osteoporosis drug is identified.

### Discovery small-molecule inhibitors of sRANKL

An in-house library of 10,016 compounds were virtually screened by docking the compounds into the above-mentioned binding site in

mRANKL (Fig. 1c) and sRANKL (Fig. 1d). This docking campaign consists of two steps:

(1)  The library was screened with the rigid-body docking packages of MOE 2018 (Chemical Computing Group, Montreal, Canada) and Glide 2018-01 (Schrödinger, New York, USA). 2,061 hits were predicted by both programs as stronger binders to sRANKL rather than mRANKL;

(2)  The hits from the step (1) were further screened with the induced-fit docking package of MOE 2018. Thus, 51 refined hits were

predicted as even stronger binders to sRANKL rather than to mRANKL.

The 51 refined hits were then validated with osteoclastogenesis assay and surface plasmon resonance (SPR) assays[34]. These result in a sRANKL inhibitor **S3** with most potent osteoclast inhibition effect ($IC_{50} = 0.096 \mu M$, KD = 34.80 μM, Fig. 2a, b and Supplementary Table 1).

To further improve the potency and selectivity of **S3**, 21 derivatives were synthesized and evaluated (Supplementary Tables 1, 2

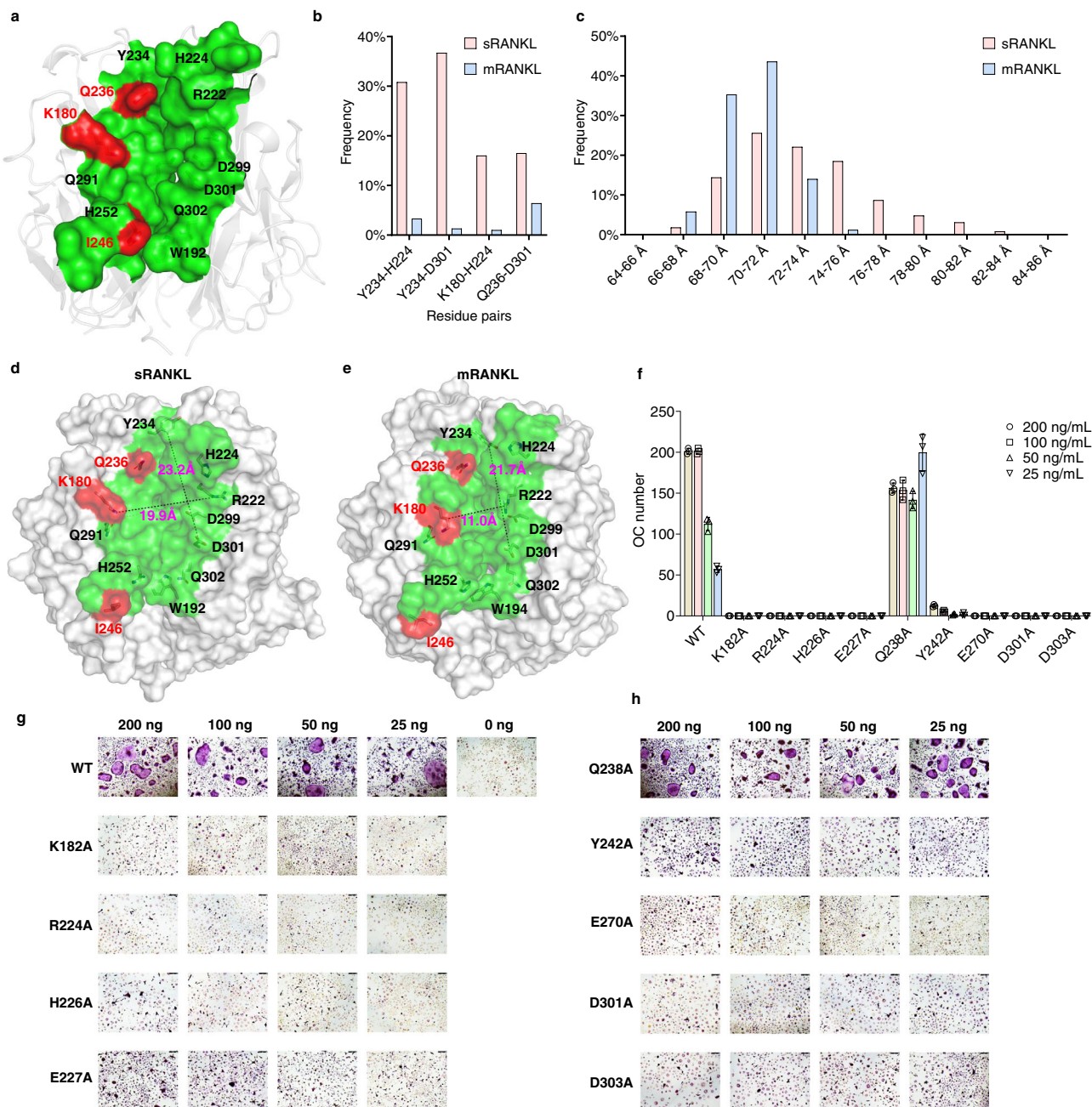

**Fig. 1 | Evidence for selective sRANKL inhibition binding site identification.** **a** Proposed small molecule binding site that can be used to distinguish sRANKL from mRANKL. **b** he probabilities that distances of residue pairs greater than 1.8 Å in sRANKL (red) and mRANKL (blue). The data are calculated based on conformation trajectories from 100 ns MD simulations on sRANKL and mRANKL. **c** SDRP frequency in sRANKL and mRANKL. **d**, **e** The distance of Y236-D303 and K182-R224 in sRANKL and mRANKL after MD simulation. **f**, **g** and **h** Osteoclastogenesis

function of single-point mutated rat sRANKL and WT sRANKL. Representative TRAP staining images from one well each, with triplicate repeats wells of BMM cells isolated from C57BL/6 mice cultured with colony-stimulating factor 1 (M-CSF (10 ng/mL) and either varying WT sRANKL or mutated sRANKL (25–200 ng/mL). The number of osteoclasts counted in the osteoclastogenesis assay with mutated sRANKL. Data were presented as group mean ± s.d., $n = 3$ biological replicates.

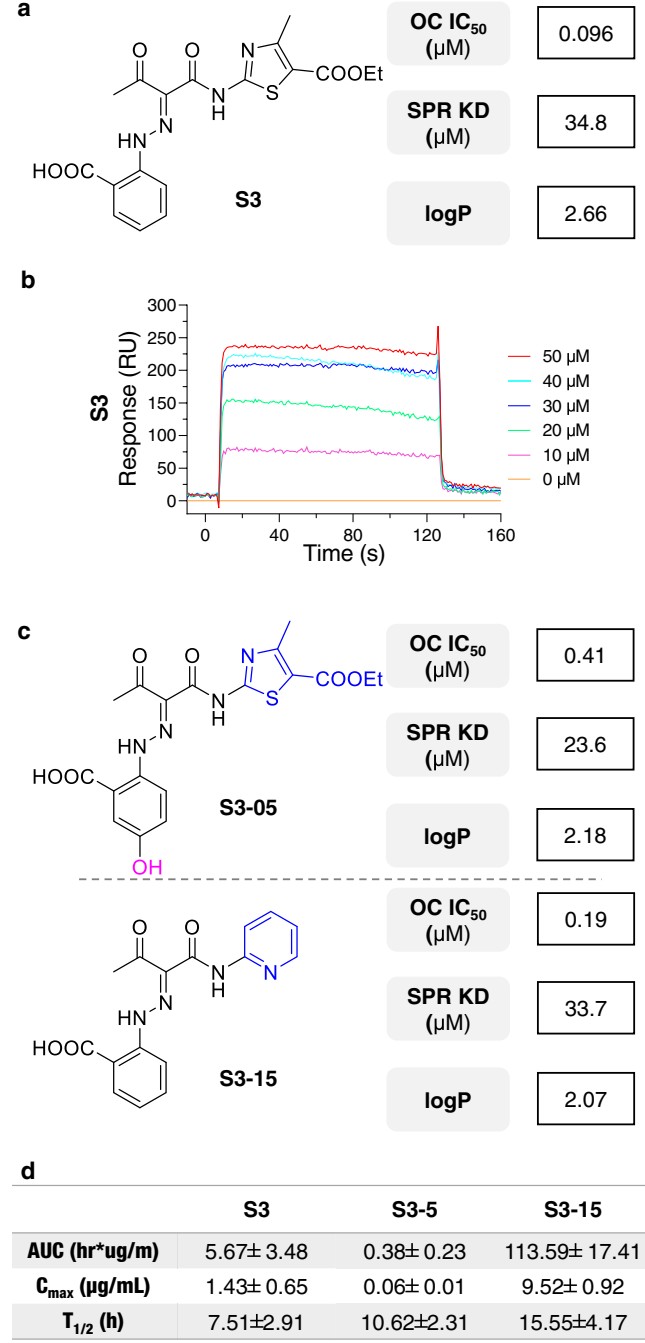

**a**

| | |
|---|---|
| OC IC$_{50}$ (µM) | 0.096 |
| SPR KD (µM) | 34.8 |
| logP | 2.66 |

**b**

**c**

S3-05

| | |
|---|---|
| OC IC$_{50}$ (µM) | 0.41 |
| SPR KD (µM) | 23.6 |
| logP | 2.18 |

S3-15

| | |
|---|---|
| OC IC$_{50}$ (µM) | 0.19 |
| SPR KD (µM) | 33.7 |
| logP | 2.07 |

**d**

| | S3 | S3-5 | S3-15 |
|---|---|---|---|
| AUC (hr*ug/m) | 5.67± 3.48 | 0.38± 0.23 | 113.59± 17.41 |
| C$_{max}$ (µg/mL) | 1.43± 0.65 | 0.06± 0.01 | 9.52± 0.92 |
| T$_{1/2}$ (h) | 7.51±2.91 | 10.62±2.31 | 15.55±4.17 |

**Fig. 2 | Selective sRANKL inhibitor discovery and structure optimization. a** The **S3** chemical structure, activity, and binding affinity against sRANKL. OC: osteoclast; SPR: surface plasmon resonance. **b** The SPR curve of compound S3 and sRANKL complex. **c** Activity, binding affinity and solubility of **S3-05** and **S3-15**. **d** The pharmacokinetic parameters of compound **S3** and newly designed compounds **S3-05** and **S3-15**.

and Fig. 2a1-4). The structure activity relationship (SAR) study was also explored (Supplementary Fig. 2b). Owing to sRANKL exists in circulation, inhibitors need to be hydrophilic that as little as possible is absorbed by cells. Therefore, compounds **S3-05** and **S3-15** (Fig. 2c) with strong osteoclastogenesis inhibition effect, potent sRANKL binding affinity and good solubility (lower logP value) were selected. A further in vivo Pharmacokinetic (PK) studies indicate **S3-15** has significant oral availability (Fig. 2d and Supplementary Fig. 2c). Hence, **S3-15** was selected for further druggability studies.

## Validating S3-15 selectivity and specificity

To further study the binding selectivity of **S3-15** to sRANKL, we linked two soluble RANKLs (termed as binary-RANKL or L-RANKL) to mimic mRANKL behavior. The binary-RANKL motif is rigidified while still maintains osteoclast differentiation ability (Supplementary Fig. 3a). Then, **S3-15** binding affinities with sRANKL and binary-RANKL were measured using isothermal titration calorimetry (ITC). The results (Fig. 3a) show that the binding affinity of **S3-15** with sRANKL (KD = 5.78 µM) is significantly stronger than the one of **S3-15** with binary-RANKL (KD = 124 µM) indicating **S3-15** selectively binds sRANKL (SI = 21). The affinities of **S3-05** binding sRANKL and binary-RANKL were also measured in ITC experiments. The results show that **S3-05** also selectively binds to sRANKL (Supplementary Fig. 3a, Table 2).

To cross-validate the above experimental results, the $^1$H saturation transfer difference (STD)-nuclear magnetic resonance (NMR) assays were used to measure the binding selectivity of **S3-15** to sRANKL and binary-RANKL. The results demonstrate that **S3-15** binds to sRANKL (Fig. 3d, red color) rather than mRANKL (Fig. 3b, green color).

As shown in Fig. 3c, the osteoclasts are significantly inhibited by the **S3-15**-sRANKL binding in the dose-dependent manner. However, the osteoclasts are weakly inhibited (IC$_{50}$ = 4.14 µM) due to the weak binding of **S3-15** with mRANKL (Fig. 3d). Interestingly, **S3-15** binds to sRANKL with a modest binding affinity, however, the potency against osteoclastogenesis is high. In this case, the simultaneously binding of sRANKL trimer with three RANK receptors are necessary for the formation of a functional sRANKL-RANK signal transduction complex to induce the osteoclastgenesis. It means that binding of a molecule of **S3-15** to anyone of the three surface clefts of sRANKL functionally blocks all the three sRANKL-RANK binding sites on sRANKL trimer, thus **S3-15**-sRANKL interaction cause a greater effect on sRANKL-RANK interactions as well as a greater biological consequence.

To further study **S3-15** inhibits osteoclastogenesis through specially binds to sRANKL, we synthesized the probes by labeling **S3** derivatives with biotin (Supplementary Fig. 3c) for activity-based protein profiling (ABPP) assay. This resulted in **S3B** and **S3-15B** (Supplementary Fig. 3c, d) for fishing protein targets. **S3B** was used as negative control because it significantly lost osteoclast inhibition (IC$_{50}$ = 49.29 µM); and **S3-15B** was used as positive control due to its similar activity as **S3-15** (IC$_{50}$ = 0.26 µM, **S3-15** IC$_{50}$ = 0.37 µM). A pulldown assay reveals that **S3-15B** can bind with sRANKL, however **S3B** cannot bind with sRANKL (Fig. 3e). In osteoclasts, both extracellular cytokines and intracellular proteins regulate osteoclastogenesis[39,40]. Therefore, three protein samples were applied to ABPP assay: (1) in vitro osteoclastogenesis culture supernatants (Fig. 3f), (2) total protein lysate of osteoclasts at the differentiation stage (Supplementary Fig. 3e), and (3) rat serum (Supplementary Fig. 3f). Several bands that specially bound to **S3-15B** were analyzed by Mass spectra, and sRANKL is observed in all three samples (Supplementary Tables 3–5). The results indicate that S3-15 binds to sRANKL more preferred than other osteoclastogenesis targets.

Two in vivo rescue experiments were conducted to further prove the osteoclastogenesis inhibition effect of **S3-15** is caused by inhibiting sRANKL. Q238A sRANKL (equivalent to hRANKL Q237A) can stimulate osteoclast differentiation (Fig. 1f, h) but without binding affinity with **S3-15** (Fig. 4a, b). Our experiments demonstrated **S3-15** can block osteoclastogenesis when the osteoclasts are incubated with WT RANKL; however, it cannot block osteoclastogenesis when incubated with Q238A sRANKL (Fig. 3g). Another H226A-mutated-RANKL (equivalent to hRANKL H225A) that cannot stimulate osteoclast differentiation (Fig. 1f, g) but with binding affinity with **S3-15** (Fig. 4a, c) was then applied. As shown in Fig. 3h, WT RANKL induced osteoclastogenesis was decreased by **S3-15**; however, it was rescued in a dose-dependent manner by adding H226A sRANKL. Using W194A sRANKL (similar as H226A sRANKL, no osteoclastogenesis activity but binds

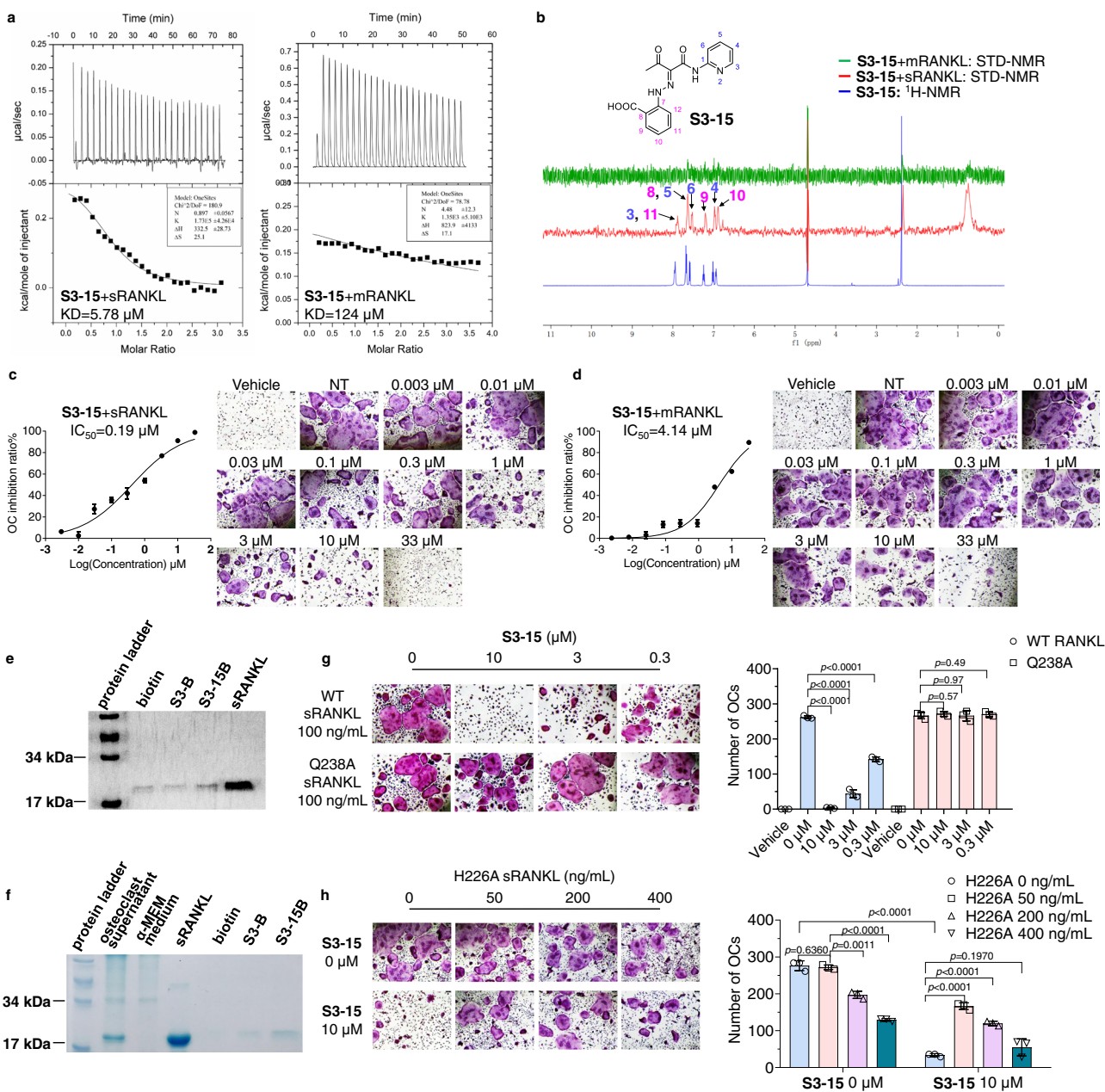

**Fig. 3 | S3-15 selectivity binding to sRANKL. a** Binding of **S3-15** to sRANKL or L-RANKL as measured by ITC. **b** Ligand-observed 1D $^1$H-NMR of the aromatic peaks of **S3-15** (blue) shows chemical shift changes in buffer. $^1$H-NMR STD transfer experiments show the appearance of aromatic peaks of **S3-15** in the presence of sRANKL (red color), while the aromatic peaks disappear in the presence of L-RANKL (green) because of the low binding affinity of **S3-15** to mRANKL. In the structure of **S3-15**, the $^1$H-NMR aromatic peaks locate in 3, 4, 5, 6 position in the pyridine ring and 8, 9, 10, 11 position in the benzene ring. **c, d S3-15** inhibits sRANKL induced osteoclast differentiation, while exhibits weak osteoclast differentiation in mRANKL induced osteoclastogenesis assay. The picture shows the IC$_{50}$ curve and a representative picture of each concentration. Scale bars, 75 μm. **e** Pull down assays

show that **S3-15B** can bind to sRANKL, while **S3B** showed no binding with sRANKL. **~f S3-15B** selectively bind to sRANKL over other proteins in supernatant of osteoclast differentiation culture. **g S3-15** is failure to bind sRANKL Q238A mutant, and the osteoclastogenesis is not inhibited. **h** sRANKL H226A mutant can partially bind with **S3-15** but cannot induce osteoclast, **S3-15** still able to inhibit osteoclastogenesis even **S3-15** is partially occupied when stimulated with 100 ng/mL WT sRANKL and different concentrition of H226A sRANKL. (Left, OC TRAP-stained pictures taken from one of three biological replicates, scale bars, 75 μm. Right, the OC numbers were counted in each well and presented in mean± s.d.) Statistical difference was determined by unpaired two-tailed Student's t test. Significant difference *p* value is < 0.05.

with **S3-15**) in the rescue experiments also have the same results (Supplementary Fig. 4a, b).

## Mechanism of selectivity studied at molecular level

To prove the binding site and its specificity, we elucidated the mechanism of **S3-15** selectively blocking sRANKL and inhibiting osteoclastogenesis through in silico and in vivo experiments. Biotin-mediated pull-down experiments with **S3-15B** were conducted with

mutated rat sRANKL proteins. The results demonstrate that **S3-15** loses binding affinity when the key residues (such as K182A, Q238A, E270A, D301A, and D305A) are mutated (Fig. 4a); and **S3-15B** still binds to the site when other residues of sRANKL (such as R192A, W194A, H254A, N255A and K283A) are mutated (Fig. 4a). STD-NMR experiments further confirm that the key residues K182, Q238, E270, D301 and D305 (Fig. 4b) are essential for **S3-15** binding, and residues R224, H226, E227, and Y242 have no contributions to **S3-15** binding to sRANKL (Fig. 4c).

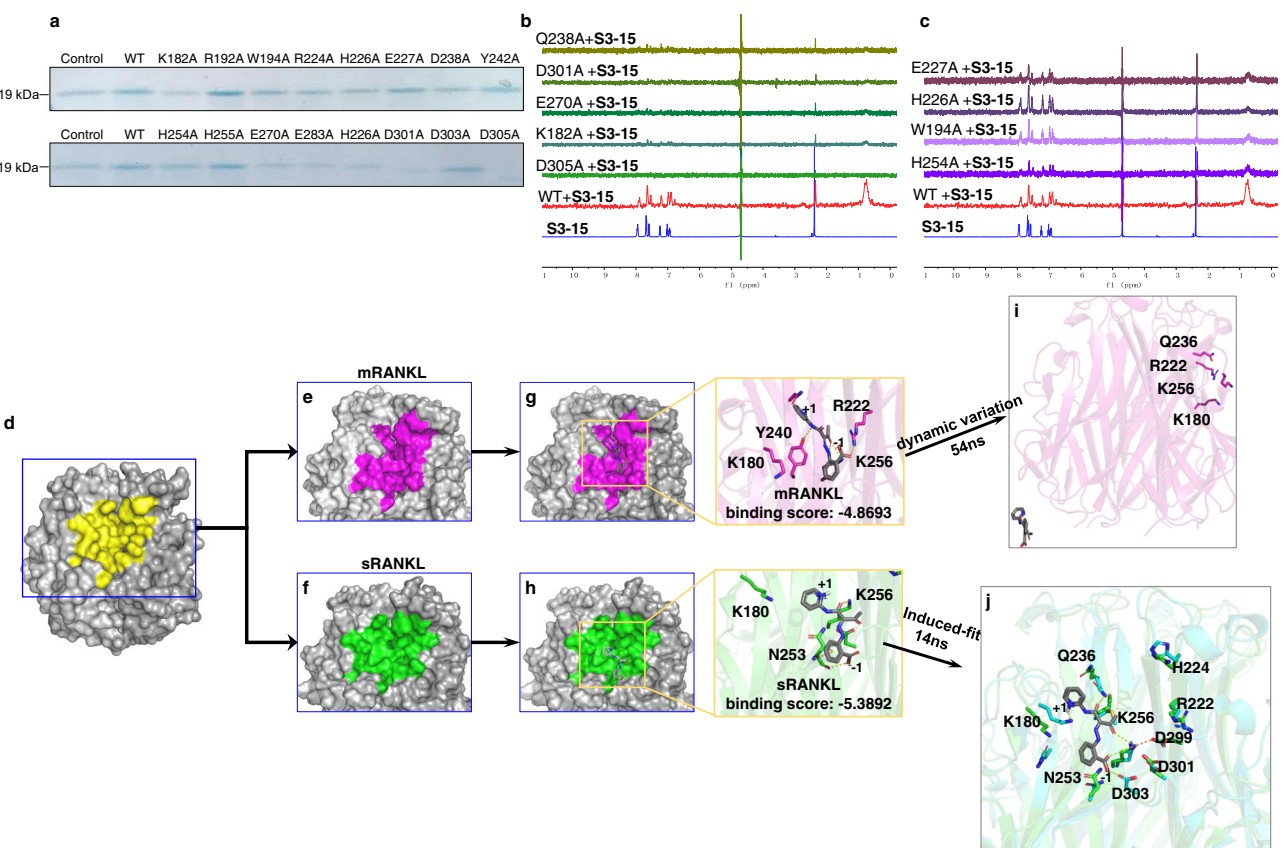

**Fig. 4 | Binding models identification of selective sRANKL inhibition. a** The pull-down assay results showing the binding affinities of **S3-15** and rat sRANKL mutants. The pull-down proteins by S3-15 were observed by Coomassie blue staining. A dimmed band means the interaction of **S3-15** and some mutants are weakened. **b** ¹H-NMR STD transfer were not observed spectral shift, indicating Q238, D301, E270, K182, and D305 sesidues are key residues of rat sRANKL to the interaction of **S3-15** and sRANKL. **c** ¹H-NMR STD transfer were observed, indicating E227, H226, W194, and H254 do not contribute to the interaction of **S3-15** and of rat sRANKL. **d** The binding surface (yellow) of sRANKL (PDB ID: 1S55). **e, f** The binding surface of mRANKL (purple) and sRANKL (green). The purple area is shrunken (compared

with the yellow area) due to the residues near by the membrane are fixed. The green area is expanded (compared with the yellow area) because sRANKL has no residues restricted by membrane. **g** MD simulations show that the complex of **S3-15**-mRANKL is not stable. **S3-15** was expelled from the binding site (purple area) after 54 ns MD simulations. **h** The complex of **S3-15**-sRANKL becomes more stable after 14 ns MD simulations (cyan color in right photo). **S3-15** has stronger binding in the binding site (green area) due to the induced-fit. H-bonds are presented as yellow dashed lines. **i** The overview of **S3-15**-mRANKL complex after 54ns MD simulations. **j** The overlay of S3-15-sRANKL after 14ns MD simulations and sRANKL.

To investigate the binding mode of **S3-15** and sRANKL, molecular docking and MD studies have been conducted. As our described, the small-molecular binding region (Fig. 4d, yellow color) is shrunken in mRANKL (Fig. 4e, purple color) and enlarged in sRANKL (Fig. 4f, green color). Thus, **S3-15** can bind to sRANKL at the enlarged stable region (binding site), and unable to bind to mRANKL at the shrunken stable region (Fig.4g, h). Actually, **S3-15** was dissociated from mRANKL after the 54th ns MD simulation (Fig. 4g and Supplementary Fig. 4c). However, **S3-15** was observed to generate induced-fit conformations that let **S3-15** interacts with K180 and Q236 (mouse RANKL K180 and Q236 are equivalent to rat RANKL K182 and Q238) and form a H-bond between **S3-15** and D303 (mouse RANKL D303 is equivalent to rat RANKL D305) (Fig. 4h, cyan color). Therefore, **S3-15** can selectively form a stable complex with sRANKL at the interface of a sRANKL dimer. This interface consists of positively charged portion (Fig. 5a, blue color) and negatively charged portion (Fig. 5a, colored in red). **S3-15** resides at the positively charged portion (Fig. 5a) because the surface of **S3-15** is mainly positively charged.

To further confirm key residues for the binding of **S3-15** and sRANKL, a docking study was applied to other **S3** derivatives. The results indicate that residues K180, Q236 and K256 in mouse RANKL are essential for inhibiting osteoclastogenesis (these residues are equivalent to rat sRANNKL K182, Q238 and K258, Fig. 5b–d and

Supplementary Fig. 4). When compounds interact with all key residues, such as **S3**, **S3-05**, **S3-08**, **S3-01**, and **S3-03**, their IC$_{50}$ values are less than 0.1 μM (Fig. 5b, Supplementary Fig. 4d–g). While the compounds (**S3-10**, **S3-00**, **S3-11** and **S3-02**) interact with one or two of the residues (mouse sRANKL K180, Q236 and K256), the RANKL binding affinities and osteoclastogenesis inhibitory activities are reduced (IC$_{50}$ > 1 μM) (Fig. 4c, Supplementary Fig. 5h–j). When compounds (such as **S3-07** and **S3-04**) do not interact with sRANKL or interact with only one of the key residues (mouse sRANKL K180, Q236 or K256), they lose osteoclastogenesis inhibitory activities (Fig. 5d and Supplementary Fig. 4k).

There three key residues provide the most important interactions with H125 and E126 in RANK for RANK-RANKL binding[34]. Our inhibitors, such as **S3-15**, occupied the binding region of H125 and E126 in RANK, and destroyed RANKL-RANK binding (Fig. 5e). This is consistent with the previous observation[38]. The GST-pull down assay was then applied with a GST tag recombinant rat sRANKL (GST-sRANKL) and its receptor RANK. As shown in Fig. 4f, the binding affinities of the **S3**, **S3-05**, **S3-07**, **S3-08**, and **S3-15** were consistent with their inhibitory activities on sRANKL-induced osteoclast formation. Further cross-link assay results indicate that **S3-15** cannot affect the trimerization of sRANKL (Supplementary Fig. 5l). These results confirmed that our sRANKL selective

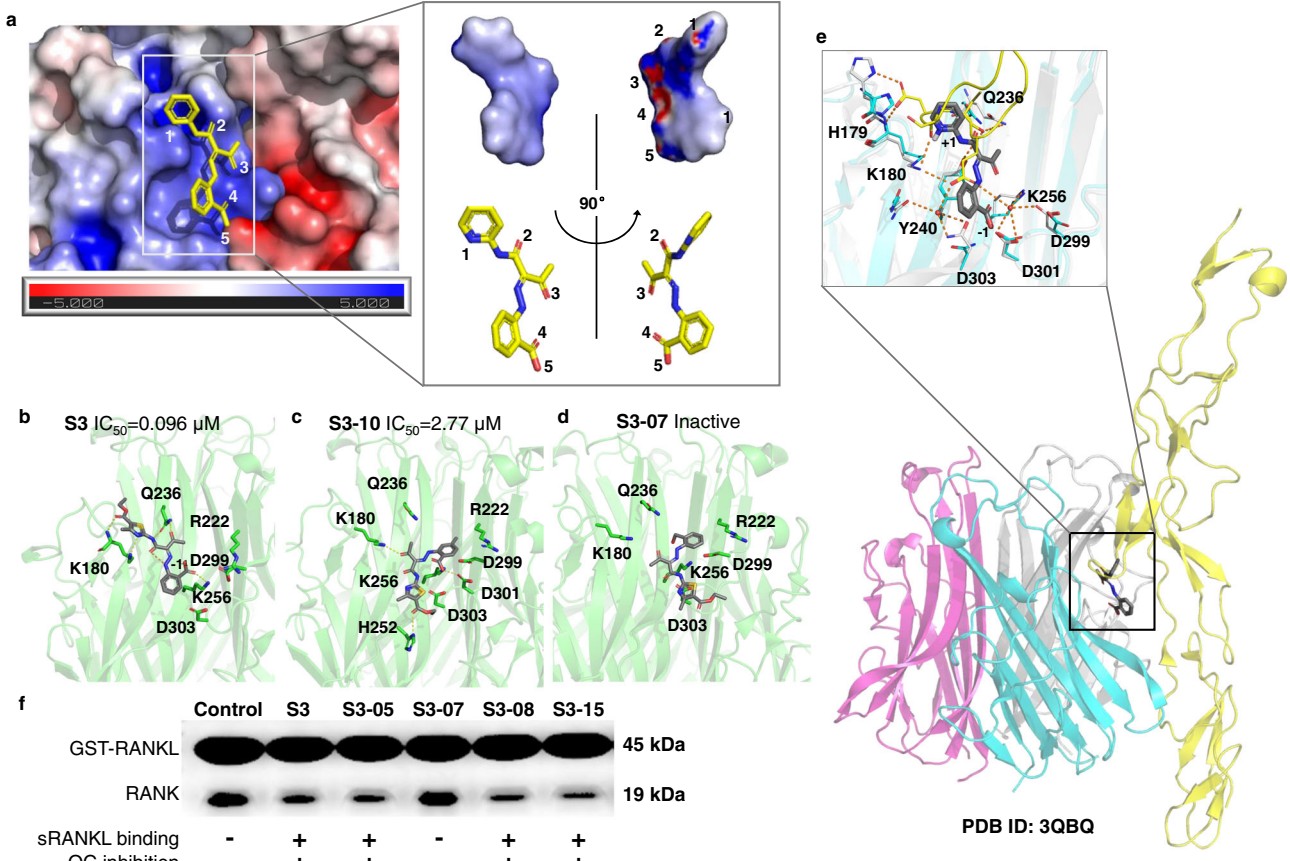

**Fig. 5 | Molecular basis of selective sRANKL inhibition. a** Binding model of **S3-15** and sRANKL: The binding site is formed at the intersection of two sRANKL monomers that are attracted by opposite static charges (blue: positive charge; red: negative charge). **S3-15** resides at the positively charged region due to **S3-15** is a negatively charged molecule (the partially negative charged atoms are labeled with 1~5). The partial changed surface was calculated with Adaptive Poisson-Boltzmann Solver (APBS) electrostatics method. **b–d** Represent three binding models of **S3**-sRANKL, **S3-10**-sRANKL, **S3-07**-sRANKL. Their binding powers are in descending order. Correspondingly, their osteoclastogenesis inhibitory activities are also in descending order. **e** The binding complex of **S3-15**-RANKL-RANK was built based on (PDB ID: 3QBQ). This model shows that **S3-15** is right in the middle of RANKL (colored in cyan) and RANK (colored in yellow) and blocks the binding of RANKL and RANK. The residues surrounding **S3-15** are the key residues for RANKL and RANK binding. **f** GST pull-down experimental results show that **S3** derivatives (**S3**, **S3-05**, **S3-07**, **S3-08**, and **S3-15**) disturb RANKL-RANK interaction leading to OC inhibition by binding to sRANKL.

inhibitors disturbed RANKL-RANK interaction without affect RANKL trimerization.

Osteoprotegerin (OPG) is a soluble decoy receptor of RANKL and prevents RANKL from binding and activating to RANK. Then, the influence of **S3-15** on sRANKL-OPG and mRANKL-OPG were evaluated. A pull-down assay data demonstrated that **S3-15** significantly suppressed sRANKL and OPG binding in high concentration (10 and 1 μM) but not lower concentration (0.3 μM) (Supplementary Fig. 5m). However, **S3-15** didn't affect on mRANKL and OPG binding (Supplementary Fig. 5n–p).

**In vitro efficacy of the selective sRANKL inhibitor (S3-15)**
To confirm **S3-15** does block the downstream signaling pathways, we have examined the effects of **S3-15** on the canonical RANKL–RANK, MAPK, and PI3K-AKT signaling pathways with Western-Blot experiments. The results revealed that **S3-15** suppresses RANKL-induced NF-κB signaling by decreasing the phosphorylation of the IκB kinase-α (IKKα), NF-κB inhibitor-α (IκBα) and p65 (Fig. 6a). It also abrogated MAPK signaling by decreasing the phosphorylation of ERK, JNK, and p38 (Fig. 6b), reduced PI3K-AKT signaling by decreasing the phosphorylation of PI3K and AKT (Supplementary Fig. 5a). Moreover, **S3-15** dose-dependently blocked both downstream transcription factor NF-κB and NFATC luciferase reporter–gene expression of RANKL-RANK signaling (Fig. 6c, d). Correspondingly, the osteoclastogenesis marker

genes, *DC-stamp*, *Ctsk*, *MMP9*, *Tracp*, *Oscar*, and *Calcr* mRNA level of were also significantly suppressed (Fig. 6e).

Nonetheless, we found that **S3-15** cannot affect TNF-α, IL-β and LPS activating NF-κB and NFATC osteoclastogenesis signaling pathway (Supplementary Fig. 3g, h). Moreover, **S3-15** cannot affect the osteoclastogenesis related gene expression without RANKL (Supplementary Fig. 3g). All the data demonstrate that **S3-15** specifically binds to sRANKL and plays a key role in osteoclast differentiation inhibition.

NF-κB and MAPK signaling pathways are related to cell survival and functions. Therefore, we examined **S3-15** in osteoclast survival and bone resorption. Apoptosis of mature osteoclasts were significantly increased after **S3-15** treatment with EC$_{50}$ of 0.55 μM (Fig. 6f and Supplementary Fig. 5b). Moreover, **S3-15** attenuated bone resorption in a dose-dependent manner (Fig. 6g). Similar effects were observed in other **S3** derivatives, such as **S3**, **S3-03**, and **S3-05** (Supplementary Fig. 5c, d).

Recent studies reveal that RANKL-RANK signaling regulates osteoblastogenesis by RANKL reverse signaling[41,42]. Therefore, the influence of **S3** derivatives on osteoblasts were examined. sRANKL was observed to significantly suppress the mineralization of osteoblast. However, **S3-15** exhibits potent activity of improving osteoblastic mineralization on human primary osteoblasts in presence of sRANKL (Fig. 6h). Another study indicates that **S3-15**, **S3** and **S3-05** increase cell proliferation and mineralization in mouse embryonic mesenchymal

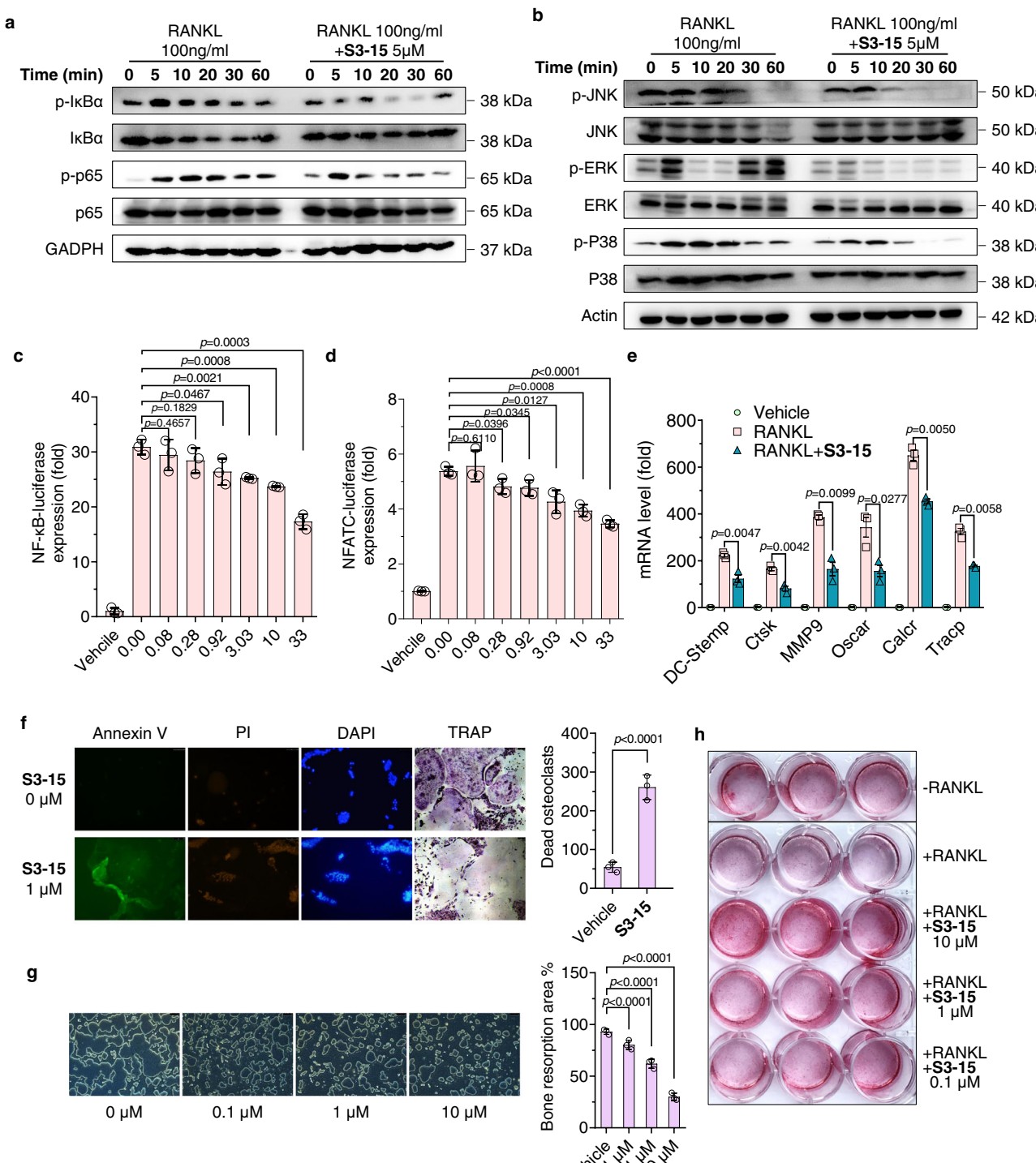

**Fig. 6 | Selective sRANKL inhibitor (S3-15) inhibits the RANKL biological functions of osteoclasts and osteoblasts. a, b** Western blot experimental results show that **S3-15** suppresses RANKL mediated NF-κB and MAPK signaling pathway. Images represent three independent experiments. **c, d** Luciferase reporter assay results show that NF-κB and NFATC-luciferase expression are reduced in doge manage when treated with **S3-15**. **e** RT-qPCR experimental results show that **S3-15** suppresses osteoclast markers (*DC-stamp, Ctsk, MMP9, Tracp, Oscar, and Calcr*) expression in BMMs cells stimulated with 100 ng/mL sRANKL. **f** Cell apoptosis assay results show that **S3-15** induces osteoclast apoptosis. The apoptosis (Annexin V stained showing green fluorescence), death (PI stained showing green fluorescence), and nucleic acid breaks (DAPI stained showing blue fluorescence

aggregation) of osteoclasts were observed. TRAP stained also observed osteoclast death. Representative images (*n* = 3 images taken in total with technical triplicate repeat) (left) and dead osteoclast (yellow arrowheads) quantification (right). **g** Bone resorption experimental results show that **S3-15** suppresses RANKL-induced resorption pits in dose-dependent manner. Representative resorption pits images (*n* = 3 images taken in total with technical triplicate repeat) (left) and resorption pits area quantification (right). **h** Human primary osteoblasts cell culture experiments show that **S3-15** promotes osteoblast mineralization. Image was taken after 12 days cell culture with **S3-15** and Alizarin Red S staining. Data are presented as the mean ± s.d. from three biological replicate per group. Statistical difference was determined by unpaired two-tailed Student's t test. Significant difference *p* value is < 0.05.

stem cell line C3H10T1/2 or human primary osteoblasts cells with osteogenic differentiation medium (Supplementary Fig. 5e, f).

## S3-15 selectively inhibits sRANKL without changing T lymphocyte differentiation

mRANKL dominates T cell differentiation and maintains dendritic cells survival and regulates T-cell/dendritic cell communication in the immune system[43]. Both mouse CD4[+] and CD8[+] T cells express mRANKL, and their proliferation are inhibited by anti-CD3Ab, anti-CD137Ab and OPG-Fc (the decoy receptor of RANKL)[44]. Hence, we used lymphocyte proliferation assay to evaluate the effect of selective inhibitor on immunity.

The results demonstrated that S3-15 did not change lymphocyte proliferation (Supplementary Fig. 6a), and cell ratio of CD4[+] T cell and CD8[+] T cell (Fig. 7a–c and Supplementary Fig. 6b). However, the nonselective RANKL monoclonal antibody - Denosumab[45] notably inhibits lymphocyte proliferation at high concentration (Supplementary Fig. 6a) together with a decreased of CD4[+] T cell differentiation (Fig. 7a–c). The immunosuppression effect of Denosumab could be due to the decreased CD4[+] ratio. In three healthy human T lymphocyte samples, we found that the proportion of CD4[+] T cells in two of them were decreased. For human 1, later stages of cell differentiation (CD4[+]CFSE[−]cells) are decreased after treated with Denosumab for 3 days (Fig. 7c, left) and 12 days (Fig. 7a, right). For human 2, after treated with Denosumab for 9 days, total CD4[+] and CD4[+]CFSE[−] T cell dose-dependently decreased, it dropped by nearly half at the concentration of 100 μg/mL (Fig. 7b). In cancer patient, the suppression effect in CD4[+] T cell seems to be more significant. After Denosumab treatment, total T cells isolated from cancer patient were reduced from $14.51 \pm 3.71\%$ to $3.88 \pm 1.77\%$ and CD4[+]CFSE[−] cell reduced from $5.32 \pm 0.76\%$ to $1.33 \pm 2.31\%$ (Fig. 7c). However, all these cells are not affected when treating sRANKL selective inhibitor S3-15 (Fig. 7a–c and Supplementary Fig. 6a–c). Those results indicated that selectively blocking sRANKL had less effect on T cell differentiation than nonselective RNAKL blockers. The immunosuppression effect of Denosumab could be mainly caused by the decreased of CD4[+] differentiation.

## Anti-osteoporosis effects for sRANKL inhibitors in vivo

Previous studies have demonstrated that the formation of osteoclasts was mainly driven by mRANKL on osteoblast, through cell-cell contact[46]. However, recent studies showed that sRANKL is sufficient to increase osteoclasts and bone resorption in vivo[47–49]. To prove that S3 series inhibits osteoporosis by selectively disturbing the binding of sRANKL-RANK, an in vivo experiment was applied. The bone volume / total volume (BV/TV) of ovariectomy (OVX) rats was increased when administrated with S3, S3-05, and S3-15 orally (Fig. 8a and Supplementary Fig. 7a–d). S3-15 can significantly improve bone trabecular parameters, and the values of BS/BV, Tb.N, and Tb.Th recovered to the sham-operated rats (Fig. 8a). In addition, in OVX mice, the trabecular parameters were also improved after treating with S3-15 (Fig. 8c). To further evaluate the biomechanical strength of the bones, a three-point bending test was applied. The result indicated that the bone strength was increased after S3-15 treatment (Fig. 8c). Moreover, the data of a histology experiment demonstrated that S3-15 significantly decreased the numbers of osteoclast as well as its potent in vivo activity (Supplementary Fig. 7f). The serum markers of bone resorption such as cross-linked carboxy-terminal telopeptide of type I collagen (CTX-I), osteocalcin, and propeptide of type I procollagen (PINP), OCN (Osteocalcin), and Glu-Ocn (Glu- osteocalcin) were significantly decreased after S3-15 administration (Fig. 8b).

Routine blood examination indicated that rats treated with S3-15 exhibited similar results to those of normal rats (sham group) and had no organ toxicity in vivo (Supplementary Fig. 7e). Similarly, S3 and

S3-05 can also increase the BMD although their efficacies are not as good as S3-15 (Supplementary Fig. 7a–d).

To prove S3-15 does not disturb the immune system in OVX-induced osteoporosis mice, the biomarkers in peripheral blood lymphocyte were examined. It was observed that the declined ratios of CD3[+] T cell and CD3[+]CD8[+] T cell, the increased ratio in CD4[+]/CD8[+] (Fig. 8d) in OVX operation mice. In CD3[+] T cell subsets, the ratio of CD4[+] T cell and CD4[+]/CD8[+] were significantly increased (Fig. 8e) in the OVX models. These results are consistent with the previous report[50]. S3-15 improved the immune condition (Fig. 8d, e), reduced the swollen spleen (Fig. 8f) in vivo. Lymphocyte transformation tests were applied to evaluate the immune function of spleen lymphocytes[51]. The lymphocyte transform rate in OVX mice were lower. OVX mice with S3-15 administration have improved transform rate that is comparable to the normal mice (Fig. 8g). In summary, the selective sRANKL inhibitor can inhibit osteoporosis without immunosuppression.

## Discussion

Discovery of small molecule PPIs inhibitors have been recognized as a major challenge due to off-target and undruggable of these small molecular inhibitors. The main reason is that the PPIs interface is a much larger and flat surface area (>800 Å²) than a conventional substrate-binding cavity and one with a less well-defined shape. Therefore, it requires the inhibitors disobey the Lipinski's rules and own large molecular weight for occupying the large region and inhibiting the PPIs. Another reason is that the classic strategy of discovering these compounds usually based on 'hot-spot' for two proteins interacting. It is reported that the hot-spot residues are usually arginine (21%) and tyrosine (12.3%). The residues surround these hot spots are hydrophobic because they can provide a shield to protect the hot spot from waters. Hence, the small molecular PPIs inhibitors usually own similar structure and large molecular weight. For these reasons, the small molecular PPIs inhibitors usually are unselective and undruggable.

Therefore, discovering druggable site on the PPIs area is preferred and it can help us to overcome the disadvantage of known PPIs inhibitors. The x-ray structure of protein, which is usually applied for drug discovery, usually presents the most stable state. However, the structure of protein is not constant, but exhibit pronounced fluctuations. This gives an opportunity to find out a traditional small molecule binding site in the surface area. Here, we perform a MD study to mimic the protein fluctuation, and thereby discovery a series druggable and selective small molecular PPIs inhibitors. These compounds not only target RANKL mainly, but also distinguish sRANKL and mRANKL which share the exact same sequence. However, our work cannot exclude that these compounds exhibit the anti-osteoporosis activity through additional mechanisms.

During the simulation, we observed that the amino acids in the surface area (rat sRANKL H226, Y236, K182 and D303) will move and explore the inside residues (rat sRANKL Y242 and K258). The fluctuations of sRANKL also provide a deeper and larger site for small molecular binding compared to x-ray structure. Further, the fluctuation difference between sRANKL and mRANKL provides us a possibility to discover sRANKL selective inhibitors. All these revealed that performing MD to mimic protein fluctuation can help us to find out the specific small molecule binding site on PPIs surface.

Here, we also show the benefit of selective sRANKL inhibitors in vitro and in vivo, which is no T-cell proliferation suppression. This hints us selective sTNFSF inhibitors overcome the side effects in clinical.

In summary, the results described herein suggest that a small molecule binding site for PPIs inhibitor discovery can be discovered by a MD study. This site is benefit for discovering more druggable inhibitors with specificity. Therefore, the future of small molecular PPIs inhibitors development will base on the protein fluctuation.

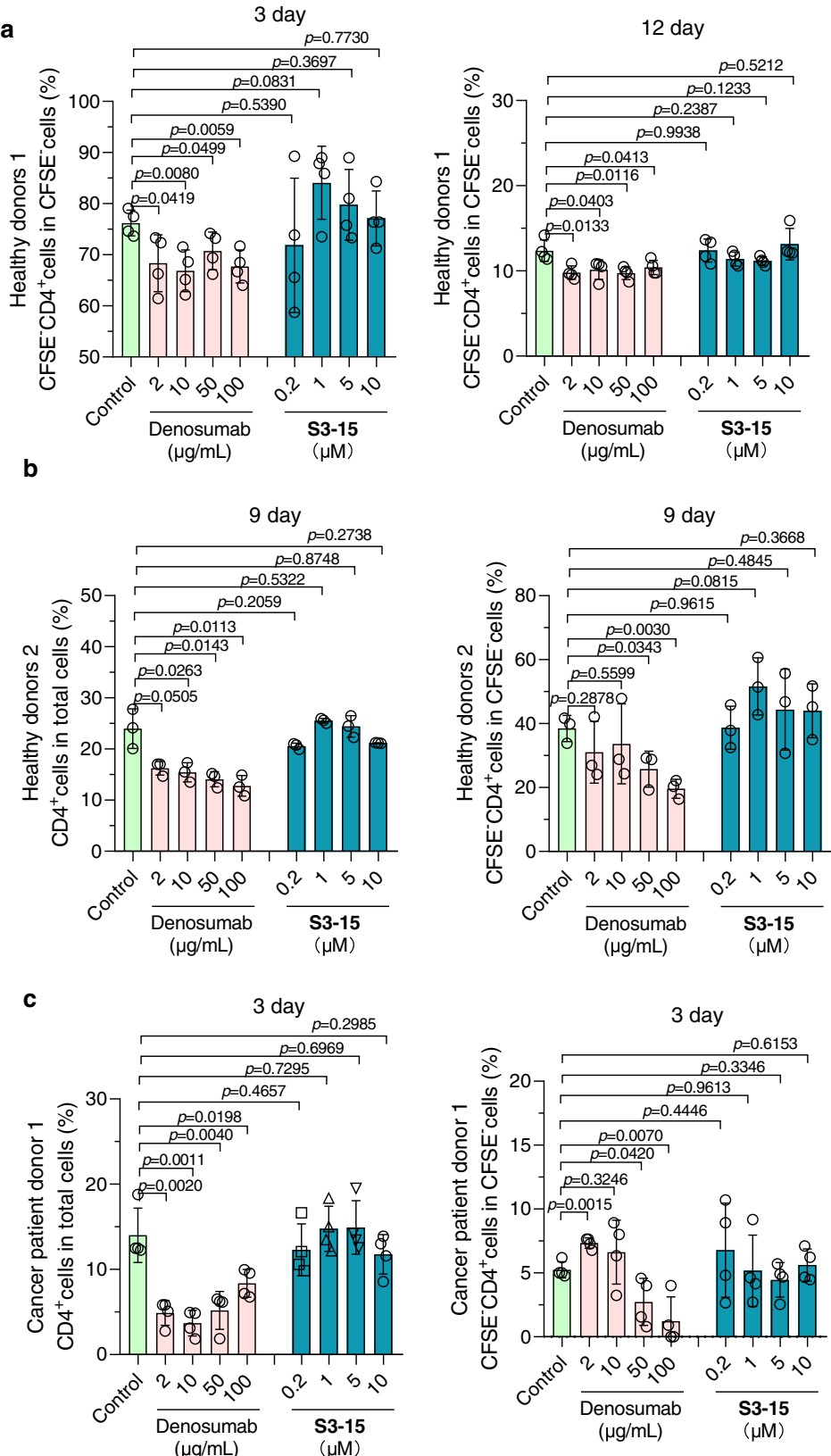

# Methods

## Materials

The information of reagents, antibodies, cell, animal, bacterial, chemicals and recombinant proteins, oligonucleotides, and software were displayed in supplemental information file name "KEY RESOURCES TABLE".

## Cell culture

All mammalian cells were culture in complete medium (containing 10% fetal bovine serum and 50 IU mL$^{-1}$ penicillin/streptomycin) and maintained in a humidified at 37 °C with 5% $CO_2$ atmosphere. RAW264.7 cells (ATCC, Cat# TIB-71) were cultured in MEM-alpha complete medium. Bone marrow cells (BMMs) were isolated from 8-week-old

**Fig. 7 | S3-15 has no adverse effects on T lymphocyte differentiation. a** CD4⁺ T cell differentiation ratio in gate of CFSE⁻ cells in T cell sample of healthy donors 1. T cells of healthy donors 1 were induced by anti-CD3 antibody (5 µg/mL) and anti-CD28 antibody (5 µg/mL) for 3 days or 12 days with/without treatment. **b** CD4⁺ T cell differentiation ratio in gate of total cells and gate of CFSE⁻ cells in T cell sample of healthy donors 2. T cells of healthy donors 2 were induced by anti-CD3 antibody (5 µg/mL) and anti-CD28 antibody (5 µg/mL) for 9 days with or without treatment. **c** CD4⁺ T cell differentiation ratio in gate of total cells and gate of CFSE⁻ cells in T cell sample of cancer patient donors 1. T cells of cancer patient donors 1 were induced by anti-CD3 antibody (5 µg/mL) and anti-CD28 antibody (5 µg/mL) for 3 days with or without treatment. In experiments of **a**–**c**, CFSE⁻ cells, CD4⁺ cells, CD8⁺ cells were assayed by flow cytometry and analyzed their percentage in the corresponding gate after treatment. Data are presented as the mean ± s.d. from three or four biological replicate per group. Statistical difference was determined by unpaired two-tailed Student's t-test. Significant difference p-value is < 0.05.

female C57BL/6 mice by flushing the marrow from the femur and tibia, then cultured in α-MEM complete medium in presence of 10 ng/mL M-CSF. After 24 h, BMMs were adherent and can be used for osteo-clastogenesis assay. RAW264.7 cells stably transfected with an NF-κB-driven or NFATc1 luciferase (kindly providing A549 cells by professor Jiake Xu in University of Western Australia) which used for reported gene assays were cultured in α-MEM complete medium. HEK293T cells (ATCC, Cat# CRL-3216) used in transfecting RANK plasmid and expressing RANK inclusion body were cultured in DMEM complete medium. All cell lines that sourced commercially or gifted from other labs were functionally validated.

### Bacterial strains
*Escherichia coli* DH5 alpha cells (Sangon Biotech, B528413-0010) were used for transforming mutated RANKL plasmids. Cells were grown at 37 °C in LB medium plates with 100 mg/mL ampicillin for plasmid construction or amplification. *Escherichia coli* BL21 (DE3) cells (Sangon Biotech, Cat#: B528414-0005) were used for protein expression for in vitro and cellular studies. Cells were grown at 37 °C in LB medium with 100 mg/mL ampicillin for plasmid maintenance.

### Animal ethics statement and care
All the animal experiments reported in this manuscript were performed accordance with an Animal Care and Use Committee (License number: SYXK 2015-0059, Guangzhou, China). Protocols were approved by the Institutional Animal Care and Use Committee of Guangdong Provincial Second Hospital of Traditional Chinese Medicine (Guangdong Provincial Engineering Technology Research Institute of Traditional Chinese Medicine).

Animals were purchased from Guangdong Medical Laboratory Animal Center, Guangzhou, China and then kept in SPF laboratory animal room under the dark/light cycle (12 h:12 h), temperature (24 ± 2 °C) and humidity (50% ± 10%) conditions. All the animals were accessed to food (Guangdong Medical Laboratory Animal Center, Cat#: (2019)05073) and water ad libitum. At the end of the experiment, euthanasia was performed with cervical dislocation.

### MD simulation for seeking sRANKL selective inhibitor binding pocket
Simulations of mouse RANKL were based on crystal structure from protein data bank (PDB ID: 1S55). The water and salt ions of crystal structure were removed. Then the missing atoms, alternate geometry or other crystallographic artifacts of crystal structure was fixed by Quick-Prep function on MOE. Followed by all hydrogens were removed. The prepared protein structures were inserted into a cubic box of explicitly represented water with 0.15 M NaCl, then neutralized by removing sodium ions. Final system dimensions were approximately 81.5 x 81.5 x 81.5 Å³. The ff14SB parameter set for protein molecules and salt ions, and the TIP3P model for water. Simulations were performed on GPUs using the CUDA version of PMEMD (Particle Mesh Ewald Molecular Dynamics) in Amber 16. The whole systems were then energy-minimized for 500 steps and simulated for 50 ps at 300 K with all protein heavy atoms. Further equilibration of sRANKL was performed at 300 K without restraints on the protein for 100 ns. Equilibration of mRANKL was also performed at 300 K with restraints of Q162, P163, F164, A165 and H166 for each monomer at 50 kcal/mol/Å² for 100 ns.

### Docking and virtual screening
The structure of RANKL was prepared from MD results analysis. A protein pose on 70 ns MD simulation was extract as sRANKL structure, and a protein pose on 26 ns MD simulation was extracted as mRANKL structure. The waters and ions on the structure were removed and then prepared by QuickPrep function on MOE.

The small-molecular structures were minimized (within RMS gradient of 0.1 kcal/mol/Å²) in MMFF94x force filed. Further, Triangle Matcher method (30 s allowed for each ligand placement) with rigid-body approach was applied to small-molecular compounds. Ten key residues were selected as docking site (K180, Y234, Q236, M238 and K256 in 1st monomer of RANKL; R222, H224, E268, D299 and D303 in 2nd monomer of RANKL). London dG scoring method was performed to evaluate the results. The compounds with S-score lower than -6 for sRANKL and mRANKL were selected independently. Among them, compounds that only docked with sRANKL were selected for further GLIDE docking.

3D structure of sRANKL were prepared by Schrödinger package (v2018-1): a) adding hydrogen atoms; b) adjusting the ionization and tautomerization state of the protein using PROPKA; c) interactive optimization of the hydrogen bonding network; and d) restrained minimization of the structure (within RMSD of 0.3 Å of the initial structure) with the OPLS3 force field. A grid box of 20.0 × 20.0 × 20.0 Å³ were formed based on same residues as MOE docking. Then a rigid-body docking was performed using GLIDE for the selected compounds. After analysis of results, the compounds with score less than −3.5. Next, these compounds were applied to induced-fit docking (IFD) by Triangle Matcher method (225 s allowed for each ligand placement).

### In vitro cytotoxicity assay
MTT assay was used to determine cell viability. RAW264.7 cells in the logarithmic growth phase were cultured in 96-well plates with $6 \times 10^3$ cells in each well and incubated for 24 h. The surviving cells were then determined by MTT method. Absorbance was measured at 570 nm using a microplate reader and displayed with OD value. Cell growth inhibition ratio was calculated as follows:

$$Cell\,growth\,inhibition\,ratio(\%) = \frac{OD\,vehicle - OD\,treaed}{OD\,vehicle} \times 100\%$$

Cell cytotoxicity $CC_{50}$ value were conducted by using *Nonlinear regression (curve fit)* analytic procedure in GraphPad Prism 8.

### In vitro anti-osteoclastogenesis activity determination
Mouse osteoclasts were cultured according to the previously reported protocols[52]. BMMs were plated in 96-well plates at a density of $6 \times 10^3$ cells/well with complete medium containing 10 ng/mL M-CSF. The following day, cells were then stimulated with 100 ng/mL sRANKL in the presence or absence of compounds with different concentrations every 2 days until osteoclasts formed. In order to calculate $IC_{50}$, we set 9 concentrations for each compound and untreated and non-sRANKL unstimulating group as controls, each with triplicated repeated wells. After 5 days, cells were fixed with 4% paraformaldehyde and then applied TRAP-staining to identify osteoclasts by using the TRAP-staining kit. Osteoclasts number (OC No.) in each well was counted and

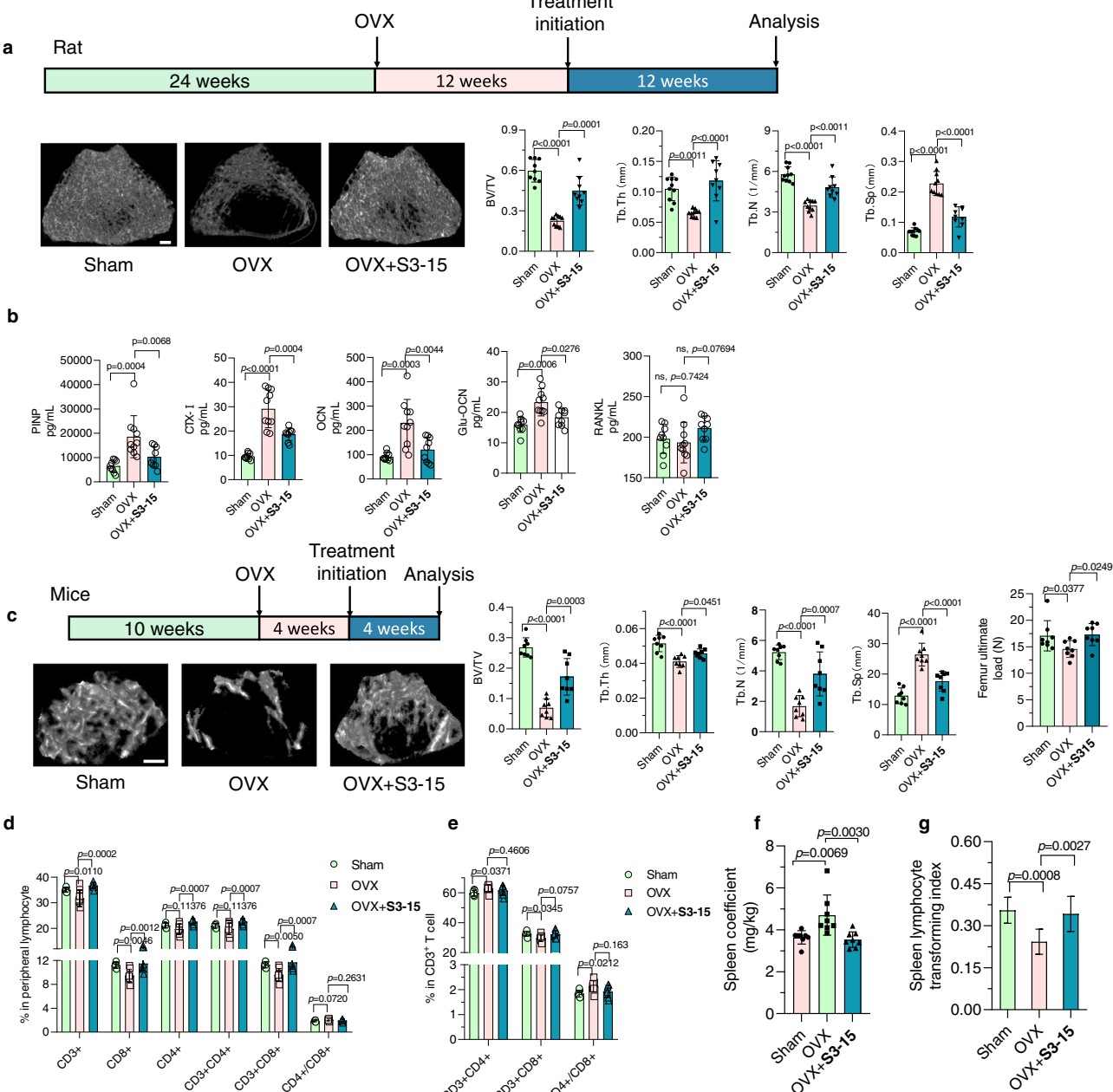

**Fig. 8 | In vivo anti-osteoporosis activity and immunological effects of S3-15.**
**a** micro-CT images of the trabecular part of distal femur of rat treated with **S3-15**
shows higher bone density than rat not treated (OVX). Histological analysis of
trabecular bone parameters including BV/TV, Tb.Th, Tb.N and Tb.Sp demonstrate
that **S3-15** improve significantly the morphometric characteristics of trabecular
bone. **b** Serum PINP, CTX-I, OCN, and Glu-Ocn osteoporosis related marker levels
were decreased in rats treated with **S3-15**. **c** micro-CT images of the trabecular part
of distal femur of mouse treated with **S3-15** shows higher bone density than mouse
not treated (OVX). Histological analysis of trabecular bone parameters including
BV/TV, Tb.Th, Tb.N, and Tb.Sp demonstrate that **S3-15** improve significantly the
morphometric characteristics of trabecular bone. Biomechanical testing shows **S3-
15** treated mice improving ultimate load. **d**, **e** **S3-15** can improve the imbalance of
immune cell ratio caused by osteoporosis. In peripheral blood total CD3+, CD8+ and
CD3+CD8+ T cells are decreased in osteoporosis mice, (left). In CD3+ T cells subset,

osteoporosis mice show an increased percentage of CD3+CD4+ T cells and a
decreased percentage of CD3 + CD8 + T cells, that results in the increased the value
of CD3+CD4+/ CD3+CD8+(right). **S3-15** treatment can reverse those abnormal indi-
cators. **f** **S3-15** alleviating spleens swelling in mice with osteoporosis. **g** Spleens of
sham, OVX and **S3-15** treated mice are got and applied splenic lymphocyte trans-
formation experiment. Spleen lymphocyte transforming index is lower in osteo-
porosis mice than sham mice and can improve to normal level after **S3-15** treatment
for 4 weeks. Rat number of sham and OVX + S3-15 group, $n = 9$. Rat number of OVX
group, $n = 10$. In mouse study, $n = 8$ mice per group. Data are expressed as
means ± s.d. Statistical difference for ultimate load was determined by one-way
ANOVA followed by Uncorrected Fisher's LSD, and the others were followed by
Tukey's multiple comparisons test. Significant difference $p$-value is < 0.05. Scalar
bar, 500 μm.

the percentage of inhibition ratio was calculated as follows:

$$OC\ inhibitory\ ratio(\%) = 100 - \frac{OC\ No.treated - OC\ No.unstimulating}{OC\ No.untreated - OC\ No.unstimulating} \times 100\%$$

$$(1)$$

The dosages and their correspondent osteoclastogenesis inhibi-
tory activity curve and $IC_{50}$ value were conducted by using *Nonlinear
regression (curve fit)* analytic procedure in GraphPad Prism 8. Photo-
graphs of individual wells were taken using Leica microscope.

## Osteoclasts apoptotic activity assay

Mature osteoclasts were generated according to "Primary cultures of BMMs and osteoclastogenesis assay". Mature osteoclasts were treated with or without compounds for 24 h[53]. At the end of incubation, cells were fixed with 4% paraformaldehyde and applied TRAP-staining. Apoptosis and cell death was measured by in situ fluorescence staining with AnnexinV-FITC (Beyotime Biotechnology) and prodidium iodide or DAPI staining flowing by their instructions. TRAP-positive cells, Annexin V, PI, and DAPI were visualized by Leica fluorescence microscopic.

## Hydroxyapatite resorption assay

BMMs ($1 \times 10^5$ cells/well) were cultured onto 6-well collagen-coated plates and stimulated with 100 ng/mL sRANKL and 10 ng/mL M-CSF for 3 days to generate early stage of osteoclasts began differentiation. Then early-stage osteoclasts were gently harvested using cell dissociation solution, and seeded into OsteoAssay Surface (hydroxyapatite-coated) 24-well plates. Early-stage osteoclasts were continue incubated in a medium containing sRANKL and M-CSF with or without treatment. After 48 h, wells were bleached to remove cells, followed by image acquisition for the measurement of resorbed areas using a Leica microscope. The percentage of surface resorbed was analyzed using ImageJ software.

## Cloning, expression, and purification of recombinant wild-type and mutant sRANKL

GST-sRANKL was expressed and purified following a previously reported protocol[36]. Rat sRANKL (aa160–318) cDNA was cloned into the bacterial expression vectors pGEX-3X and pGEX-2T with purification GST tag and resultant plasmid named p3rRANKL1. Label-free sRANKL protein in this study was obtained by removing His$_6$-sumo tag from fusion protein His$_6$-sumo-sRANKL. The DNA sequence encoding sRNAKL was inserted downstream the His6-sumo tag of a modified pET20b (+) plasmid. Single point mutations of sRANKL (K182A, R192A, W194A, R224A, H226A, E227A, Q238A, Y242A, H254A, N255A, E270A, K283A, D301A, D303A, or D305A) were constructed by PCR amplification using primer pairs to introduce the designed mutations to pET20b-His6-sumo-sRANKL recombinant plasmid. L-RANKL (binary-RANKL) plasmid was constructed as follow: the consecutive DNA sequence sequentially encoding His6-sumo tag and two sRANKL monomers was cloned into pET20b by PCR amplification using primer pairs to introduce the following restriction enzyme sites to the coding sequences: Nde I-His6-sumo-sRANKL-Bam HI and Bam HI-sRNAKL-Xho I. Each PCR product was double digested with the appropriate restriction enzymes (Thermo) and ligated to the pET20b vector between the Nde I and Xho I sites. Gly-Ser-Gly-Ser was designed to link the two monomers. After confirmed by sequence analysis, the recombinant plasmids were transformed into *Escherichia coli* BL21 (DE3) cells. The transformed cells were grown in LB media till OD$_{600}$ about 0.6, then 0.15 mM IPTG was added to induce protein overexpression for 16 h at 20 °C.

The subsequent purification steps were carried out at 4 °C. Cells were harvested and resuspended in lysis buffer, followed by ultrasonic disruption cell for 30 min. To purify the proteins, the cleared supernatant of GST or His lable proteins were loaded onto glutathione sepharose or Ni-NTA column and washed with lysis buffer. Then GST-sRANKL fusion protein was eluted with reduced glutathione elution buffer. 0.5 mg Ulp1 was added to Ni-NTA column and incubated in 4 °C overnight, then sRANKL was released from sumo-sRANKL by Ulp1 enzymatic hydrolysis.

GST-sRANKL was further purified by a Superdex 200 Increase 10/300 G (GE Healthcare) column in Tris-HCl buffer (10 mM Tris-HCl pH 8.0, 20 mM NaCl). sRANKL, mutational sRANKL, and L-RANKL were further purified by HiLoad 16/60 Superdex 200 pg (GE Healthcare) column in buffer containing 200 mM NaCl, 20 mM Tris, PH8.0. Finally, the pure proteins were concentrated 20~25 mg/mL. Proteins were quantified using the BCA method by using BCA Protein Assay Kit (Thermo Fisher Scientific) and storage at −80 °C.

The sequences of sRANKL and mRANKL mimics were seen in Supplementary Information.

## Cloning, expression, and purification of recombinant RANK

The methods of expression and purification of recombinant RANK are similar to previous reports[54,34]. DNA sequence encoding mouse extracellular RNAK fragment (residues 26–210) was cloned into pET28a using Nde I and Xho I restriction enzyme sites. The recombinant RANK was produced as inclusion body in *E. coil* Bl21(DE3) cells. The transformed cells were grown in LB in LB media till OD$_{600}$ about 0.6, 0.5 mM IPTG was added to induce protein overexpression in 37 °C for 7 h. The cell pellet harvest after centrifugation was resuspended in washing buffer containing 50 mM Tris pH 8.0, 150 mM NaCl, 5 mM EDTA and 1% Triton X-100. The cells were disrupted by sonication for 20 min (2 s on, 2 s off). After repeating the sonication and centrifugation, the precipitate was dissolved in 6 M guanidine hydrochloride, 50 mM Tris (pH 8.0), 1 mM EDTA, 150 mM NaCl, and 10 mM DTT to a protein concentration of ~30 mg/mL with stirring at room temperature for 2 hr. The soluble RANK was diluted to ~10 mg/mL using buffer containing 20 mM Na$_2$HPO$_4$ pH 7.3, 1 M L-arginine, 20% glycerol, 10 mM reduced glutathione and 1 mM oxidized glutathione. Then, sequential dialysis was followed to refold protein, against 20 mM Na$_2$HPO$_4$ pH 7.3, 0.5 M L-arginine, and 10% glycerol for 12 h, 20 mM Na2HPO4 pH 7.3, 0.2 M L-arginine, and 5% glycerol for 12 h, and finally twice against 20 mM Na2HPO4 pH 7.3 for 12 h. After centrifugation, the refold RANK was concentrated and further purified by Superdex 200 Increase 10/300 GL (GE Healthcare).

## Surface plasmon resonance (SPR) assay

A ProteOn XPR36TM SPR instrument (BioRad Hercules, CA) was used for carrying out SPR assays. sRANKL protein in PBS buffer, pH 7.4, was diluted to 20 nM in 10 mM sodium acetate buffer, pH 5.5 and immobilized in parallelflow channels of a the EDAC/Sulfo-NHS activated surface of a GLH biosensor chip (Bio-Rad) with an amine coupling kit (Bio-Rad). The surface was blocked with 1 M ethanolamine. The final immobilization level for sRANKL was approximately 14,000 RU. A blank immobilization channel in the same GLH biosensor chip was also activated with EDAC/Sulfo-NHS and blocked with 1 M ethanolamine to use as a reference.

Biosensor binding experiments were conducted in filtered and degassed running buffer (PBS containing 0.005% Tween-20, pH 7.4) at 25 °C. Compounds was dissolved in 100% DMSO in appropriate different series concentrations of stock solutions. Before detecting, compounds were diluted in six or five concentrations with running buffer and ensured that the content of DMSO contained in each detected concentration was consistent. Compounds in different concentrations were injected simultaneously at a flow rate at 20 mL/min for 180 s and 120 s for the association phase, followed by 180 s for the disassociation. Following the compound injection, the chip surface was regenerated with 40 s pulses of 0.85% H3PO4 and running buffer.

ProtedOn Manager 2.0. was used for data analysis. Each set of sensorgrams was globally analyzed using the 1:1 Langmuir binding model to obtain the kinetic rate constants (Kon and Koff). Global kinetic rate constants (ka and kd) were derived for each reaction, and the equilibrium dissociation constant, KD, was calculated using the equation KD = kd/ka.

## STD NMR binding assay

For NMR studies, purified sRANKL or L-RANKL protein samples were buffer exchanged into PBS (Gbico, C10010500BT, pH7.4) using a Millipore spin column ($1500 \times g$ for 15 min at 4 °C, repeated six times, each time adding fresh PBS and discarding the flow-through). Protein

concentration was determined using BCA method according to the procedure of Pierce™ BCA Protein Assay Kit (Thermo Fisher Scientific). Compound **S3-15** powder was dissolved in $H_2O$ as a concentration of 10 mM.

Protein saturation transfer difference (STD) experiments were performed on Bruker Avance III 500 (500 MHz) spectrometer (Columbia University) at 300 K using triple-resonance cryogenic probes optimized for proton detection. All experiments were performed using an independent sample for each experimental measurement as a 500 μL sample in NMR tubes with 100 μM protein and 5 mM **S3-15** in final concentration. Uniformly, for STD assay all samples were contained 50 μL protein, 200 μL **S3-15** (10 mM in H2O), and 250 μL $D_2O$; for **S3-15** two-dimensional spectra reference, samples were 50 μL PBS, 200 μL **S3-15** (10 mM in H2O) and 250 μL $D_2O$.

## Isothermal titration calorimetry (ITC) assay

ITC titrations were performed on the MicroCal iTC200 system at 25 °C[55]. Working stocks of compound **S3-15** were prepared in 100% DMSO at 50 mM and then diluted to 500 μM at PBS with final DMSO concentration at 0.8% (v/v). PBS with 0.8% DMSO was used as ITC buffer. Purified sRANKL or L-RANKL protein was buffer exchanged into same ITC buffer using a Millipore spin column. **S3-15** at 500 μM was loaded into a syringe and protein was loaded into an isothermal calorimeter cell. During titration, **S3-15** in the cell was injected (2 μL per injection, 25 injections in total, and 180 s between each injection) regularly from a rotating syringe into protein solution in the isothermal calorimeter cell. The sample cell was stirred at 385 rpm. Reference cell power was set to 5 μcal/s. Control experiments were conducted under the same conditions using buffer solution instead of **S3-15**. Data were analyzed using a one-site binding model in Origin 7.1 software for ITC version 7.0 (MicroCal). The dissociation constant, KD, was calculated according to equation KD = 1/K. All other parameters, K (varying the stoichiometry), N (varying the stoichiometry), ΔH (enthalpy of the reaction), and ΔG (entropy of the reaction) were determined from the titration data.

## GST pull-down assay

The RANKL-RANK interaction destructive activity of sRANKL selective inhibitors were detected by GST pull-down assay. 120 μg purified GST-sRANKL in 1 mL PBS containing 1% BSA buffer was loaded into the resin and incubated at room temperature for 45 min. After that, compounds **S3**, **S3-05**, **S3-07**, **S3-08** and **S3-15** with final concentration of 50 μM were added and incubated for 30 min. And then put 25 μg RANK into each tube, gently mix, incubated for another 20 min. 1% BSA PBS buffer with 0.8%(v/v) DMSO solution was used as control tube and performed with the same procedure. The resins were subsequently centrifugal settling and washed with 1 mL PBS 3 times and resuspended in 1xSDS gel loading buffer (20 μL PBS, 5 μL 5 × Loading Buffer) and denatured at 100 °C for 5 min. The denatured samples were analyzed by SDS-PAGE electrophoresed. The gel was immersed in Coomassie blue staining solution and eluted in destaining solution on the shaker overnight. Finally, the band were observed, photographed and analysis of grayscale by Image J.

## WT sRANKL or mutation sRANKL enrichment by pull-down assay

500 μM **S3-15**-biotin conjugate (**S3-15B**, positive control), biotin (B, blank control), and/or **S3**-biotin conjugate (**S3B**, negative control) in 400 μL PBS buffer were incubated with 40 μL streptavidin bead suspension for overnight at 4 °C. After conjugated, the beads were collected and washed with PBS buffer thrice, and then resuspended in 400 μL PBS buffer with 100 μg WT sRANKL or 15 mutation sRANKL proteins in each tube. All the tubes were incubated with stirring at 4 °C for 1 h. The beads were gently washed with 500 mL of PBS buffer for three times. The beads were resuspended in 20 μL PBS buffer with 5 μL

5× Loading Buffer and denatured at 100 °C for 5 min. The denatured samples were analyzed by SDS-PAGE electrophoresed and Coomassie blue staining. Finally, the band were observed, photographed and analysis of grayscale by Image J.

## In vitro pull-down assay of sRANKL or mRANKL binding to OPG

To evaluate whether **S3-15** affects the binding of sRANKL or mRANKL to OPG, we did the pull-down assay.

sRANKL-OPG: 200 ng human sRANKL with his-sumo tag was incubated with Ni gel at 4 °C for 10 min. Then the Ni gel was washed with PBS for 3 times. 500 μL PBS that contains 100 ng Fc tag human OPG. Then **S3-15** was treated in different concentration (0, 10, 1 and 0.3 μM). The mixtures were incubating at 4 °C for 30 min. Washed with PBS for 3 times. 500 μL 1% Triton X-100 in PBS was then added at 4 °C. The concentration of OPG was tested by OPG ELISA kit (USCN, China).

mRANKL-OPG: MC3T3-E1 cells were differentiated by osteoblast differentiation medium (Pythonbio, China) for 2 days. Then cells were exposed to 1 M NaCl for 5 minutes to remove endogenous OPG, washed once in PBS, and fixed with 70% ethanol for 5 minutes at 4 °C. Fixed cells were wash once by PBS and then exposed to PBS that containing 100 ng mouse OPG (Sino, China) or 100 ng mouse OPG with 1 μM **S3-15** and incubated for an additional 30 min at 4 °C. Cell layers were washed 2 times with PBS and lysed with 1% triton in PBS. The concentration of OPG in cell lysates were measured with an OPG ELISA kit (USCN, China). Similar proceed were performed using HEK 293 T cells that transfer full length mouse RANKL or human RANKL.

## Affinity-based target validation with cell lysates, culture supernatant or rat serum

Target enrichment was performed by incubating early-stage OC cell lysates, culture supernatant or serum samples with streptavidin beads pre-loaded with biotinylated compound **S3-15B** (positive control), B (blank control), and **S3B** (negative control). To obtain cell lysates and culture supernatant, BMMs were cultured and stimulated osteoclastogenesis for 3 days. Cell culture supernatant was collected, and cells were by PBS and lysate by 1× RIPA with phosphatase inhibitors and EDTA-free protease inhibitors. Then we collected the cell lysates and centrifuged at 18,386 × g for 10 min at 4 °C. To obtain rat serum sample, a female rat serum was extracted form aortaventralis after anesthesia, follow by centrifuged at 18,386 × g for 10 min at 4 °C. All the protein samples were adjusted to 1 mg/mL with PBS buffer and incubated with streptavidin beads pre-loaded with biotinylated compounds at 4 °C for 1 h with sharking. Then the beads were washed, denatured and separated proteins by SDS- PAGE electrophoresis. Finally, the band in gels were revealed by Coomassie blue staining or silver staining. In-gel protein digestion and desalted of differential bands in the gels were performed as described[56] with some modification. Briefly, the gel slices were hydrated with 100 μL of 50 mMNH4HCO3:30%CH3CN (v:v) and incubated at room temperature for 6 h. This hydrated step was only performed to gel with Coomassie blue staining but not silver staining. Hydrated solvent was eliminated and all the gel slices form both Coomassie blue staining and silver staining were dried with 300 μL 100%ACN Gel slices followed by shocking 5 min and then freeze-drying 3 min. Dried slice were then rehydrated with 300 μL of 10 mM DTT/ 50 mM NH4HCO3 and incubated at 56 °C for 60 min. Remove solvent and repeated the dried gel procedure once more. All samples were added 300 μL of 60 mM IAA/ 50 mM NH4HCO3 and left in dark at room temperature for 30 min. Removed IAA solution and dried the gel with 300 μL 100%ACN by 5 min sharking and 3 min freeze-drying. Then added 80 μL 50 mM NH4HCO3 and 8 μL 0.25 μL /uL trypsin solution (dissolved in 10 mM acetic acid). The digestion was carried out at 37 °C overnight. Tryptic peptide extracts with 200 μL 0.1% FA acetonitrile by sharking for 5 min and speed vac to dry. The dry digested samples were desalted using C18 Stage Tip (Thermo Fisher) according to the manufacturer's

instructions. The desalted samples were sent to School of Pharmaceutical Sciences Central Laboratory in Sun Yat-sen University and performed the LC-MS/MS protein identification by nanoRPLC-Q Exactive Orbitrap (Thermo Fisher).

## Reporter-gene assay for NF-κB and NFAT

In order to investigate the NF-κB and NFAT transcriptional inhibition activity of selective sRANKL inhibitor, luciferase reporter gene assays were used[57]. RAW 264.7 cells stably transfected with an NF-κB luciferase reporter gene (3κB-Luc-SV40) or NFAT luciferase reporter gene were seeded into 96-well plates at a density of $1.5 \times 10^5$ or $1.0 \times 10^5$ cell per well respectively for 24 h. Cells were pretreated with compounds for 1 h, and then stimulated with sRANKL (100 ng/mL), TNF-α (20 ng/mL), IL-β (50 ng/mL) and LPS (500 ng/mL) for 6 h for NF-κB transcriptional or 24 h for NFAT transcriptional. After treatment both cells were lysed and measured luciferase activity using a Promega Luciferase Assay system (Promega Corporation) by a Varioskan flash Multimode Microplate Reader (Thermo Scientific).

## RT−qPCR and Western Blotting

The mRNA expression level of **S3-15** was measured by RT−qPCR as primary described[52]. Total cellular RNA sample was prepared using RNAiso Plus, followed by reverse transcription with PrimeScript™ MixRT Master Mix reverse transcriptase kit. Then quantitative real-time PCR assays were performed with TB Green™ premix EX Taq™ II kit on an Applied Biosystems. RANKL-RANKL signaling inhibiting effect was determined by western blotting. RAW264.7 cells were treated with 5 μM **S3-15** with or without sRANKL (100 ng/mL) for 5, 10, 30 and 60 min after starved for 1 h. Then the cells were washed with cold PBS and lysed by RIPA Lysis Buffer. Cell lysates were then processed for Western Blotting analyses. Primary antibody binding was detected using rat secondary antibodies (1:5000) coupled with enhanced chemiluminescence (ECL) reagents (Thermo Scientific) and visualized on Tanon 5200. Primers used for qPCR and antibodies used for western blotting were seen supplementary materials for details.

## Rescue experiments

For the osteoclast formation induced by WT and Q238A sRANKL experiments, BMMs were seeded in 96-well plate in density of $6 \times 10^3$ cell/ well with α-MEM completed medium containing 10 ng/mL M-CSF for 24 h. Then the cells were treated with 0 μM, 0.3 μM, 3 μM and 10 μM and stimulated with 100 ng/mLWT sRANKL or Q238A sRANKL respectively. For the osteoclast formation induced by WT, W194A, and H226A sRANKL experiments, cells were stimulated with 100 ng/mL WT sRANKL combined W194A, H226A sRANKL in different concentrations (50 ng/mL, 200 ng/mL, and 400 ng/mL) with or without 10 μM **S3-15**. After 5 days, osteoclasts were formed, and applied TRAP-staining as described above. TRAP-positive multinucleated cells with > 3 nuclei were scored as osteoclasts.

## T lymphocyte cell proliferation assay

T lymphocyte cell proliferation assay was performed according to a reported method[58] with some modification. Briefly, $5 \times 10^6$ cells /mL cells were stained 2.5 μM CFSE (Tonbo Biosciences) and incubated at 37 °C for 12 min. Then washed and discarded the supernatant and adjusted the CFSE stained cell to $1.75 \times 10^6$ cells /mL. Cells were diluted in 1640 completed medium containing 5 μg/mL anti-CD3 antibody (Tonbo Biosciences) to $3 \times 10^5$ cells/mL and seed in a 96-well plate or 24-well plate that pre-packaged 0.4 μg/well anti-CD28 antibody and 1.6 μg anti-CD28 antibody (24-well) (Tonbo Biosciences). Cells were then cultured for 3 days, 9 days or 12 days. Subsequently, the proportion of CD4+ and CD8+ cells was determined by flow cytometry (Beckman Coulter, FC500) according to manufacturer's instruction of PE-Cyanine7 anti-Human CD8a and PE anti-Human CD4 antibodies (Tonbo Biosciences). Data analyzing was performed by using CXP software (Beckman Coulter, Supplementary Fig. 8).

## Splenic lymphocyte transformation test

Spleens were aseptically isolated from sham group, OVX group and OVX treatment group after treating **S3-15** for 4 weeks in a clean bench (II Class biosafety Cabinets, Escolifesciences, Singapore). Splenic single cell suspensions were obtained by gentle squeezing spleen tissue by injecting 2 mL cold Hank's solution. Splenic cells were filtered through 40 μm nylon filters (BD Biosciences, Darmstadt, Germany), and centrifuged at $1400 \times g$ for 10 min. Then cells were resuspended in 1640 completed medium (containing 10%FBS, 1% penicillin/streptomycin, 1% glutamine (200 mmol/L) and $5 \times 10^{-5}$ mol/L 2-mercaptoethanol) with a concentration of $5 \times 10^6$ cells/mL and seeded 100 μL into 96-well plates. Two wells for each mouse, and one well stimulated with 50 μL CoA (Sigma-Aldrich®, Germany) solution (working concentration 7.5 μg/mL in 1640 completed medium), another well added 50 μL 1640 complete medium as a control. After that, the plates were incubated at 37 °C in a $CO_2$ incubator at 5% $CO_2$ for 72 h. After stimulation, cells were measured by MTT method. Stimulation of cell proliferation ability was determined as the absorbance (OD) value of stimulated minus unstimulated cells for each cell sample.

## Pharmacokinetic studies in rat

28-week-old and 280 g~370 g weight male SD (Sprague Dawley) rats were used for pharmacokinetic (PK) studies. Rats were weighed and divided into groups of 3 rats per group. Compounds **S3**, **S3-05** and **S3-15** were dissolved in deionized water and made the concentration to 1 mg/mL. Then rats were oral gavage administrated at the doses of 10 mg/kg for each compound. Blood samples were collected in tubes with heparin sodium at 5 min, 10 min, 20 min, 30 min, 40 min, 60 min, 2 h, 3 h, 4 h, 5.5 h, 7 h, 9 h, 12 h, 16 h, 24 h, 32 h, 40 h, 48 h, time points, with three mice per time point for each compound. Blood samples were centrifuged $2100 \times g$ for 10 min at 4 °C, then the plasma was transferred to a clean microfuge tube. Plasma samples were stored at −80 °C. Additionally, ten rats were used as a control and not treated with compound. Plasma from control mice was used to calibrate LC-MS and correct for matrix effects. Drug concentration in plasma was quantified by LC-MS-based methods. Pharmacokinetic parameters were estimated using Phoenix WinNonlin (version 6.3) (Certara USA, Princeton, New Jersey) from mean plasma concentration-time profiles.

## Osteoporosis therapeutic study in rats and mice

Female SD rats or C57BL/6 mice were cared until they grown up to 6 months old or 10 weeks old, respectively. Then we subjected them to either bilateral ovariectomy or sham operation after anesthesia with 10% chloral hydrate (0.3 mL/100 g). Then all the rats or mice were intramuscular injected of penicillin to prevent infection in the first week of surgery. For rat experiments, after 12 weeks of operation, 3 rats from sham-operated or ovariectomized groups were random selected to determine bone mass by micro-CT imaging biopsy to verify successful produced bone loss(data unshown). Then the OVX rats were individually administrated with **S3-15**, **S3**, and **S3-05** orally with (10 mg/kg/day) for another 12 weeks. For mouse experiments, after 4 weeks of surgery, the OVX mice were treated with 10 mg/kg/day **S3-15** for another 4 weeks.

The weight of rats or mice were recorded every week in the treatment period. After treatment, the rats were harvested, the serum, heart, liver, spleen, lung, kidney, thymus, and femurs were all collected and subjected to subsequent analysis. The right femurs of rats or mice were cleaned excess soft tissue, fixed in 10% formalin, and processed them for microcomputed tomography (micro-CT) analysis. Left mice femurs were collected and excarnate for biomechanical strength

testing. Mice peripheral blood were as well as collected for flow cytometry analyzing.

## Micro-CT analyses

The right femurs of rats or mice were further analyzed with micro-CT on Inveon PET/CT (Siemens, Germany). We used micro-CT to scan the femur from the femoral head to the femoral condyle, using 19 µm resolution, 80 kV 500uA, 360 projection, 3000-6000 image threshold, full rotation cone beam. Three-dimensional reconstruction and histo-morphometric analyses of trabecular bone were conducted with the IAW (Inveon analysis workstation) analysis software. The trabecular extended 1 mm (rat) or 0.5 mm (mice) proximally to the end of the distal growth plate over 2 mm (rats) or 1 mm (mice) toward the diaphysis was selected as the region of interest (ROI) for the analysis. The resulting two-dimensional images of trabecular bone in relative cross sections were shown in grayscale. Trabecular bone parameters were measured including bone volume/tissue volume (BV/TV), bone surface area/bone volume (BS/BV), trabecular number (Tb.N), trabecular thickness (Tb.Th), trabecular separation(Tb.Sp) and trabecular pattern factor(Tb.P.F).

## Histological examination

The mice left femur was fixed in paraformaldehyde for 72 h, demineralized using 10% Ethylenediamine tetra-acetic acid (EDTA-2Na) for 3 weeks, and then dehydrated with ethanol, clarified with xylene, and embedded with paraffin. Paraffin embedded tissue was sectioned on a rotary microtome. The sectioned tissues were stained with TRAP. The histologic changes of the femur caused by the ovariectomy were observed with a light microscope.

## Osteoporosis biomarkers determination

Change of rat osteoporosis biomarkers after treating with **S3-15** were determined by enzyme-linked immunosorbent assay (ELISA). Bone resorption serum biomarkers including CTX-I (C-terminal telopeptide of type I collagen), OC(Osteocalcin), PINP (Procollagen I N-Terminal propeptide), and RANKL(Receptor activator of nuclear factor-κB ligand). Rat blood was collected from aortaventralis after anesthesia, and centrifuged in $14000 \times g$ at 4 °C to get the serum samples. The protein contents of those biomarkers in serum samples were detected according to manufacturer's instruction (CTX-I and OC, Systems IDS, UK; RANKL, R&D Systems, USA; PINP, USCN Life Science, Inc., China.).

## Serum biochemical markers determination

After treating with **S3-15** for 12 weeks, rat serum was harvested and analyzed routine blood biochemical markers, including calcium (Ca), phosphorus (P), total cholesterol (TC), triglyceride (TG), glucose (Glu), aspartate aminotransferase (AST), alanine aminotransferase (ALT), alkaline phosphatase (ALP), total bile acid (TBIL), total bilirubin (TBA), creatinine (CREA), urea nitrogen (BUN). Ca, P, AST, ALT, ALP, CREA, and BUN were purchased from Wako Pure Chemical, Japan; Glu,TC,

TG, TBIL, and TBA were purchased from Kehua Bio-engineering Co.LTD, China. Routine blood biochemical markers were examined with a Hitachi chemistry automated analyzer (Hitachi 7020, Japan) according to manufacturer's instruction.

## Rats or mice organ coefficients determination

We dissected all rats after 12 weeks of treatment, then took the main organs like heart, liver, spleen, lung, kidney, and thymus. All the organs were removed the attached adipose and weighted with an electronic balance (Sartorius, Germany). Mice spleen and cervical lymph node were also taken and weighted. Organ index (mg/g) was calculated as the following formula:

$$\text{Organ coefficients} = \frac{Organ\,weight(mg)}{Rat\,or\,mouse\,total\,weight(g)}$$

## Splenic lymphocyte transformation test

Spleens were aseptically isolated from sham group, OVX group and OVX treatment group after treating **S3-15** for 4 weeks in a clean bench (II Class biosafety Cabinets, Escolifesciences, Singapore). Splenic single cell suspensions were obtained by gentle squeezing spleen tissue by injecting 2 mL cold Hank's solution. Splenic cells were filtered through 40 µm nylon filters (BD Biosciences, Darmstadt, Germany), and centrifuged at $1400 \times g$ for 10 min. Then cells were resuspended in 1640 completed medium (containing 10%FBS, 1% penicillin/streptomycin, 1% glutamine (200 mmol/L) and $5 \times 10^{-5}$ mol/L 2-mercaptoethanol) with a concentration of $5 \times 10^6$ cells/mL and seeded 100 µL into 96-well plates. Two wells for each mouse, and one well stimulated with 50 µL CoA (Sigma-Aldrich®, Germany) solution (working concentration 7.5 µg/mL in 1640 completed medium), another well added 50 µL 1640 complete medium as a control. After that, the plates were incubated at 37 °C in a $CO_2$ incubator at 5% $CO_2$ for 72 h. After stimulation, cells were measured by MTT method. Stimulation of cell proliferation ability was determined as the absorbance (OD) value of stimulated minus unstimulated cells for each cell sample.

## Mice peripheral blood T lymphocyte subsets analysis

In order to study the effect of **S3-15** on the immune system, after 4 weeks of **S3-15** treatment, orbital venous blood with EDTA-anticoagulated was collected for T lymphocyte subsets analysis in osteoporosis mice. We used flow cytometric to determinate T lymphocyte subsets according to the guidelines for Flow Cytometric of Beckman Coulter, FC500. Cells were stained according to manufacturer's instruction of PE-Cyanine7 anti-Human CD8a and PE anti-Human CD4 antibodies (Tonbo Biosciences). Data analyzing was performed by using CXP software (Beckman Coulter, Supplementary Fig. 9).

## General procedure for compound synthesis

(1 mmol, 2 eq.). Water (3 mL) was also added, and stirred at 0 °C for 30 min to form mixture A. Compounds **2** (0.5 mmol, 1 eq.) was dissolved in MeOH (5 mL), followed by adding NaOAc (1.1 mmol, 2.2 eq.) and $H_2O$ (5 mL) to form mixture B. The mixture A was added into mixture B by drop wise, stirred at room temperature for 8 h. The precipitate was filtered, and washed with a mixture of MeOH : H2O = 3 : 1 to give **S3** series. Detail of synthetic methods were seen in "Compounds synthesis" of Supplementary Information.

Scheme S1. The synthesis procedure of **S3** series compounds.

*tert*-Butyl-3-oxobutanoate (1 mmol, 1 eq.) and substituted amine **1** (2 mmol, 2 eq.) was added into xylene, stirred at 120 °C for 12 h. Then the mixture was extracted with $CH_2Cl_2$, washed with $H_2O$ and brine. The organic layer was dried in vacuum. Purifying the mixture by flash column (EtOAc : n-Hexane = 1 : 3) provided desired compounds **2**.

Different substituted anilides **3** (0.5 mmol, 1 eq.) was dissolved in the MeOH (3 mL), followed by adding HCl (1 mmol, 2 eq.) and NaNO2

## Statistical analysis

All statistical analysis was performed using Graphpad Prism Version 8. All the in vitro or in vivo data were shown as mean ± s.d. In vitro studies n represents biological replicates, in the animal studies n represents sample size (the number of mice or rats). For SPR and ITC graphs,

individual data points are plotted and calculated binding parameters by using the analytical software of the instrument according to the points value. Dose-dependent inhibition studies were statistically analyzed using two-tailed, unpaired *t*-test. For the comparisons of multiple groups, statistical analysis method was ANOVA. A *p*-value of < 0.05 was considered to be statistically significant.

## Data availability

The synthesis routes and analytical spectra of the chemical compounds presented in this study are provided in the Supplementary Information document. The crystal structure of RANKL used in this study are available in Protein Data Bank database under accession code 1S55 and 3QBQ. The protein mass spectrometry mass spectrometry results of pull-down assay in excel form. Sequence data from this article can be found in supplementary information. Source data are provided with this paper. All data supporting the findings of this study are available from the corresponding authors upon reasonable request. Source data are provided with this paper.

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

## Acknowledgements

This work was supported by the Natural Science Foundation of China (81573310, 82104473, 22177140), GuangDong Basic and Applied Basic Research Foundation (2019A1515110484) and the Science and Technology Planning Project of Guangdong Province (2016A020217005). We thank Prof. J.X. (School of Biomedical Sciences, University of Western Australia) for kindly providing full length mouse and human mRANKL plasmids and other kindly assistance.

## Author contributions

D.H. and C.Z. contribute equally to this work. D.H. carried out most of the pharmacological and biological experiments, analyzed data, and wrote the paper. C.Z. carried out the experiments of the molecular modeling and chemical synthesis, wrote the paper. R.L. contributed to the in vitro and in vivo activity evaluation experiments. B.C. contributed to protein cloning, expression, and purification experiments. Y.Z. carried out plasmid transfection and functional evaluation experiments. Z.S. and J.K. performed the pharmacokinetic studies. Q.G. H.Z., and J.X. designed and supervised this project, as well as revised the manuscript.

## Competing interests

The authors declare no competing interests.
