## [Peer Review File · Nature Communications]

REVIEWER COMMENTS

Reviewer #1 (Remarks to the Author):

The paper entitled " Identifying Small Molecular Binding Site to Selectively Inhibit sRANKL-RANK Interactions for Anti-osteoporosis Drug Discovery" by Dane Huang is potentially interesting. In the study, the authors found that a binding site that allows a small molecule to selectively interrupt sRANKL-RANK interaction and, not to interfere mRANKL-RANK interaction due to the membrane rigidifies the mRANKL. They screened and identified a potent osteoporosis inhibitor, S3-15, that might selectively inhibit sRANKL RANK interaction by in silico and in vitro experiments. The results are interesting and novel.

There are some issues

As stated in this paper, sRANKL is a non-covalently formed homotrimer, and cleaved from mRANKL. mRANKL has C terminal extracellular connecting stalk binding to the membrane. What are the exact coding amino acid sequences of sRANKL and mRANKL that were used in this study??

What is the potency between sRANKL and mRANKL in osteoclastogenesis, a IC50 of S3-15' effect on osteoclastogenesis is only meaningful if the effects of sRANKL and mRANKL in osteoclastogenesis are similar at the basal level?

How S3-15 affects the effect of sRANKL and mRANKL on self trimerization, an important part of RANKL function?

Does S3-15 affect the binding of sRANKL and mRANKL to OPG, an important decay receptor to RANK?

Figure S7 could be supported by histology showing osteoclast parameters that have been affected?

Might need to discuss the intracellular effect of this small molecule, S3-15 that might have residue inhibitory effect on osteoclastogenesis other than its inhibitory effect on sRANKL alone?

Reviewer #2 (Remarks to the Author):

In this study, they authors reported the discovery of small-molecule inhibitors of the interactions soluble RANK ligand (sRANKL)-RANK as potential therapeutic agents for the treatment of osteoporosis. The authors hypothesized that sRANKL and membrane RANKL may exhibit different flexibility and size with respect to their binding sites, which would allow the discovery of small-molecule inhibitors that specifically inhibit the interactions of sRANKL-RANK but not of the interactions of mRANKL-RANK. MD simulations revealed that sRANKL indeed has a different size and flexibility in its binding site as compared to mRANKL. Employing docking, the authors identified putative small-molecule inhibitors for sRANKL. Surface plasmon resonance (SPR) experiments identified one such small-molecule (S3), which binds to sRANKL with a KD value of 34.80 uM and inhibits osteoclast in vitro with an IC50 value of 0.096 uM. Testing additional analogues of S3 identified compounds S3-05 and S3-15 with strong osteoclastogenesis inhibition effect, potent sRANKL binding affinity and good solubility and oral bioavailability. The authors performed extensive in vitro experiments (NMR and ITC, among other) to demonstrate that S3-15 binds to soluble RANKL. Activity-based protein profiling using a biotin-labeled S3-15 (S3-15B) showed that S3-15B binds to sRANKL more preferred than other osteoclastogenesis targets. Two in vivo rescue experiments using mutated sRANKL proteins further demonstrated that the osteoclastogenesis inhibition effect of S3-15 is caused by inhibiting sRANKL. Additional experiments further showed that those key residues in sRANKL involve in the binding with S3-15. In vitro efficacy and mechanistic studies showed that S3-15 suppresses RANKL-induced NF-kB signaling and blocks both downstream transcription factor NF-kB and NFATC luciferase reporter-gene expression of RANKL-RANKL signaling. Apoptosis of

mature osteoclasts was significantly increased after S3-15 treatment with EC50 of 0.55 μM . The authors also showed that S3-15 selectively inhibits sRANKL without changing T lymphocyte differentiation, different from Denosumab, a antibody RANKL inhibitor. Finally, the authors demonstrated anti-osteoporosis effects for S3-15 and two analogues in a rat model by oral administration. Altogether, this study provides an important proof-of-concept that targeting sRANKL-RANK interactions may be a new therapeutic approach for anti-osteoporosis drug discovery. This is an excellent and complete study and is recommended for publication in Nature Communications. A number of points should be clarified.

1. The authors have done an excellent job to demonstrate that S3 and its new analogues such as S3-15 bind to sRANKL and the anti-osteoporosis activity of S3-15 is due to blocking the interactions of sRANKL-RANK. However, it is surprising to see that S3-15 binds with sRANKL with a modest binding affinity ($K_d = 33.7 \mu\text{M}$ by SPR and $5.78 \mu\text{M}$ by ITC) could have such potent activity in biological assays (e.g. osteoclast inhibition $\text{IC}_{50} = 0.19 \mu\text{M}$). Typically such a discrepancy between a binding affinity to its proposed therapeutic target and its biological activity would suggest that the real cellular target is not what was proposed. A discussion is needed to clarify this point.

2. The authors have used cells and models from different species (rat and human) in different experiments. A clarification is needed on the sequence homology between human sRANKL and rat sRANKL.

3. From its chemical structure, S3, S3-15 and other analogues appear to be not stable. It is recommended that the authors provide stability data for these compounds in those key assays used in this study.

4. For modeling, the protonation state of those inhibitors need to be specified. For example, pyridine would be positively charged and the acid would be negatively charged.

Reviewer #3 (Remarks to the Author):

Manuscript NCOMMS-21-31271-T

Identifying Small Molecular Binding Site to Selectively Inhibit sRANK-RANK Interactions for Anti-osteoporosis Drug Discovery

A new compound, which binds to soluble RANKL (sRANKL) without binding to membrane bound RANKL (mRANKL) is created and tested in an ovariectomized (OVX) rats and mice model of osteoporosis.

Page numbering is missing from the main document, but is present in the supplemental material.

The Introduction needs to more carefully explain the need for selectively binding to sRANKL. Especially when it is taken into consideration that it has been shown that activated T cells both express mRANKL and secrete sRANKL (Immunology Letters 94, 239–246, 2004).

Page 15, first paragraph

Bone Mineral Density (BMD) is not shown in any of the figures. Please replace with BV/TV (which is highly correlated to vBMD). Moreover, OVX + S3 had not significantly higher BV/TV than OVX (Figure S7B, this relationship is marked "ns").

Page 16, Figure 7

The 3D reconstructions are too small to see anything. You can easily omit BS/BV and TBPf to make more room in the figure. "VOX" -> "OVX"

In Figure 7 C, please start the y-axis in 0 (e.g. Tb.Th).

Please compare the ability of the identified compounds to counteract OVX-induced bone loss with that of denosumab at comparable doses (similar to Figure 6).

The influence on the trabecular microstructure is interesting, but the most important property is the bone fracture strength. Please report the mechanical strength of the bones of the rats and mice.

Page 19, supplemental material

The 3D reconstructions are too small and BS/BV and TBPf can be omitted. If you want an expression of the curvature of the trabeculae, Structure Model Index (SMI) is a better option than TBPf anyway.

In Figure S7 B + C "OVX +" is missing from the x-axis title.

Page 35, micro-CT analyses

What do you mean with "After BMD detected"? A voxel size of 50 μm is much too small when the rat trabecular thickness is approximately 60 μm (Figure 7A) and the mouse trabecular thickness is approximately 40 μm for the OVX animals. According to the Nyquist sampling theorem the smallest object that can be resolved using a voxels size of 50 μm is 100 μm . This low spatial resolution may explain the look of the 3D reconstructions.

Please, state the length of the integration time used and the threshold used for image segmentation (please follow the current guidelines when reporting micro-CT findings: Journal of Bone and Mineral Research 25(7):1468–1486, 2010).

Responses to the reviewers

Reviewer #1 (Remarks to the Author):

The paper entitled “ Identifying Small Molecular Binding Site to Selectively Inhibit sRANKL-RANK Interactions for Anti-osteoporosis Drug Discovery” by Dane Huang is potentially interesting. In the study, the authors found that a binding site that allows a small molecule to selectively interrupt sRANKL-RANK interaction and, not to interfere mRANKL-RANK interaction due to the membrane rigidifies the mRANKL. They screened and identified a potent osteoporosis inhibitor, S3-15, that might selectively inhibit sRANKL RANK interaction by in silico and in vitro experiments. The results are interesting and novel.

The reviewer is highly appreciated for the positive remark and recommendation. The specific questions are answered as follows.

There are some issues

1. As stated in this paper, sRANKL is a non-covalently formed homotrimer, and cleaved from mRANKL. mRANKL has C terminal extracellular connecting stalk binding to the membrane. What are the exact coding amino acid sequences of sRANKL and mRANKL that were used in this study?

Response: Per reviewer’s suggestion, we list the exact coding amino acid sequences of sRANKL and mRANKL in “Supplemental Data” page38, at line 20. The contents are also listed here for your review:

“The amino acid sequence of sRANKL was the C-terminal extracellular region of RANKL (rat 159-318 aa). The amino acid sequence of mRANKL was two sRANKL monomers linked by -Gly-Ser-Gly-Ser-.

(1) sRANKL (rat 159-318 aa)

GKPEAQPF AHLTINAANIPSGSHKVSLSWYHDRGWAKISNMTLSNGKLRVN
QDGFYYLYANICFRHHETSGSVPADYLQLMVYVVKTSIKIPSSHNL MKGGST
KNWSGNSEFHFYSINVG GFFKLRAGEEISVQVSNP SLLDPDQDATYFGAFKVQ

DID

(2) GST-sRANKL (rat 159-318 aa)

GKPEAQPF AHLTINAANIPSGSHK VSLSSWYHDRGWAKISNMTLSNGKLRVN
QDGFYYLYANICFRHHETSGSVPADYLQLMVYVVKTSIKIPSSHNL MKGGST
KNWSGNSEFHFYSINVG GFFKLRAGEEISVQVSNPSLLDPDQDATYFGAFKVQ
DID

(3) mRANKL (sRANKL-GSGS-sRANKL, rat 159-318 aa)

GKPEAQPF AHLTINAANIPSGSHK VSLSSWYHDRGWAKISNMTLSNGKLRVN
QDGFYYLYANICFRHHETSGSVPADYLQLMVYVVKTSIKIPSSHNL MKGGST
KNWSGNSEFHFYSINVG GFFKLRAGEEISVQVSNPSLLDPDQDATYFGAFKVQ
DIDGSGSGKPEAQPF AHLTINAANIPSGSHK VSLSSWYHDRGWAKISNMTLSN
GKLRVNQDGFYYLYANICFRHHETSGSVPADYLQLMVYVVKTSIKIPSSHNL
MKGGSTKNWSGNSEFHFYSINVG GFFKLRAGEEISVQVSNPSLLDPDQDATY
FGAFKVQDID.”

2. What is the potency between sRANKL and mRANKL in osteoclastogenesis, a IC50 of S3-15' effect on osteoclastogenesis is only meaningful if the effects of sRANKL and mRANKL in osteoclastogenesis are similar at the basal level? How S3-15 affects the effect of sRANKL and mRANKL on self trimerization, an important part of RANKL function?

Response: The questions are appreciated, and answered as follows:

The effects of sRANKL and mRANKL in osteoclastogenesis are similar at the basal level indeed. However, over-expressed sRANKL induces excessive osteoclastogenesis that causes osteoporosis in patient (Acta Endocrinologica (Buc), 2009, vol. V: 27-40). We have modified the manuscript by adding a concise description on this in the page 2, at line 26 of the manuscript. The description is copied in here for your review:

Normally, the functions of sRANKL and mRANKL are similar in osteoclast differentiation. However, over-expressed sRANKL can cause excessive bone resorption that induces osteoporosis ¹⁶.

To answer the reviewer's questions, we have conducted a crosslink assay to measure

the effect of **S3-15** on RANKL self-trimerization. As shown in the following figure, we did not observe significant difference at the trimer bands between the DMSO (the control), **S3** and **S3-15** -treated sRANKL samples. The sRANKL dimer and monomer also exhibited same results. We prove that **S3-15** does not affect sRANKL trimerization process *in vitro*. These results have been documented in the page 11 at line 18 and, Figure S4L of the manuscript. The cross-link assay results indicate that **S3-15** cannot affect the trimerization of sRANKL (Figure S4L). Thus, we are convinced that our sRANKL selective inhibitors does attenuate RANKL-RANK interaction without affect RANKL trimerization.”.

Here is the Figure S4L.

3. Does S3-15 affect the binding of sRANKL and mRANKL to OPG, an important decay receptor to RANK?

Response: To answer this question, we have performed a number of experiments. The pull-down assay was conducted to explore the effects of **S3-15** on sRANKL and OPG binding. Our experiments demonstrate that **S3-15** affects the binding of human sRANKL to OPG at high concentration (10 and 1 μM) but not low concentration (0.3 μM). Three more experiments on mRANKLs have conducted to study if **S3-15** affects the binding of mRANKL to OPG. Two of the experiments are on human and mouse mRANKL expressed in 293T cell, the other is on the mRANKL expressed in MC3T3-E1 osteogenic precursor cells. The results demonstrate that **S3-15** cannot affect the binding of mRANKL to OPG at concentration of 1 μM. These results have been documented in the page 11 at line 21 and Figure S4M-S4P of the manuscript. Here is the paragraph we have added to the manuscript:

“Oteoprotegerin (OPG) is a soluble decoy receptor of RANKL and prevents

RANKL from binding and activating to RANK. Then, the influence of S3-15 on sRANKL-OPG and mRANKL-OPG were evaluated. A pull-down assay showed that S3-15 significantly suppress sRANKL and OPG binding in high concentration (10 and 1 μ M) but not lower concentration (0.3 μ M) (Figure S4M). However, S3-15 didn't affect on mRANKL and OPG binding (Figure S4N-4P).”

Here is the Figure S4N-4P.

4. Figure S7 could be supported by histology showing osteoclast parameters that have been affected?

Response: In order to answer this question, we have performed histology experiments. We have observed that **S3-15** significantly decreases the number of osteoclasts in femur compare with OVX group. These results have been documented in the page 17 at line 16 and Figure S7G of the manuscript. Here is the Figure S7G.

5. Might need to discuss the intracellular effect of this small molecule, S3-15 that might have residue inhibitory effect on osteoclastogenesis other than its inhibitory effect on sRANKL alone?

Response: To answer this question, we have generated sRANKL- Q238A mutant to evaluate the intracellular effect of **S3-15** on osteoclastogenesis. The mutant causes osteoclastogenesis without binding to **S3-15** (Figure 4B).

Based upon these results, we conclude that **S3-15** cannot inhibit osteoclastogenesis that is induced by the RANKL- Q238A mutant. This proves that **S3-15** doesn't have residue inhibitory effect on osteoclastogenesis other than its inhibitory effect on sRANKL alone (Figure 3G).

Here is the Figures 4B and 3G.

Fig. 4B

We have also conducted experiments to prove that **S3-15** doesn't have residue intracellular inhibitory effect (Figure S3H-S3J). We have found that **S3-15** cannot affect other osteoclastogenesis promoting factors (TNF- α , IL- β and LPS) induced NF- κ B and NFATC activation. Moreover, **S3-15** cannot affect the osteoclastogenesis related gene expression without RANKL either. We reveal that **S3-15** doesn't have residue inhibitory effect on osteoclastogenesis.

The results are documented in the page 9 at line 2 and Figure S3H-S3J of the manuscript. The following sentences have been added:

“Furthermore, we have found that **S3-15** cannot affect TNF- α , IL- β and LPS activating NF- κ B and NFATC osteoclastogenesis signaling pathway (Figure S3H and S3J). However, **S3-15** can significantly inhibit the activation of sRANKL on NF- κ B and NFATC pathway (Figure 5C and 5D). Moreover, **S3-15** cannot affect the osteoclastogenesis related gene expression without RANKL (Figure S3J). Nonetheless,

S3-15 significantly reduces these gene expression when RANKL is performed (Figure 5E). All the data revealed that **S3-15** specially binds to sRANKL to inhibit osteoclast differentiation.”

Here are Figures S3H-S3J.

Reviewer #2 (Remarks to the Author):

In this study, they authors reported the discovery of small-molecule inhibitors of the interactions soluble RANK ligand (sRANKL)-RANK as potential therapeutic agents for the treatment of osteoporosis. The authors hypothesized that sRANKL and membrane RANKL may exhibit different flexibility and size with respect to their binding sites, which would allow the discovery of small-molecule inhibitors that specifically inhibit the interactions of sRANKL-RANK but not of the interactions of mRANKL-RANK. MD simulations revealed that sRANKL indeed has a different size and flexibility in its binding site as compared to mRANKL. Employing docking, the authors identified putative small-molecule inhibitors for sRANKL. Surface plasmon resonance (SPR) experiments identified one such small-molecule (S3), which binds to sRANKL with a KD value of 34.80 uM and inhibits osteoclast in vitro with an IC50 value of 0.096 uM. Testing additional analogues of S3 identified compounds S3-05 and S3-15 with strong osteoclastogenesis inhibition effect, potent sRANKL binding affinity and good solubility and oral bioavailability. The authors performed extensive in vitro experiments (NMR and ITC, among other) to demonstrate that S3-15 binds to soluble RANKL. Activity-based protein profiling using a biotin-labeled S3-15 (S3-15B) showed that S3-15B binds to sRANKL more preferred than other osteoclastogenesis targets. Two in vivo rescue experiments using mutated sRANKL proteins further demonstrated that the osteoclastogenesis inhibition effect of S3-15 is caused by inhibiting sRANKL. Additional experiments further showed that those key residues in sRANKL involve in the binding with S3-15. In vitro efficacy and

mechanistic studies showed that S3-15 suppresses RANKL-induced NF- κ B signaling and blocks both downstream transcription factor NF- κ B and NFATC luciferase reporter-gene expression of RANKL-RANKL signaling. Apoptosis of mature osteoclasts was significantly increased after S3-15 treatment with EC50 of 0.55 μ M. The authors also showed that S3-15 selectively inhibits sRANKL without changing T lymphocyte differentiation, different from Denosumab, a antibody RANKL inhibitor. Finally, the authors demonstrated anti-osteoporosis effects for S3-15 and two analogues in a rat model by oral administration. Altogether, this study provides an important proof-of-concept that targeting sRANKL-RANK interactions may be a new therapeutic approach for anti-osteoporosis drug discovery. This is an excellent and complete study and is recommended for publication in Nature Communications. A number of points should be clarified.

The reviewer is highly appreciated for the positive remarks and recommendation. The specific questions are answered as follows.

1. The authors have done an excellent job to demonstrate that S3 and its new analogues such as S3-15 bind to sRANKL and the anti-osteoporosis activity of S3-15 is due to blocking the interactions of sRANKL-RANK. However, it is surprising to see that S3-15 binds with sRANKL with a modest binding affinity ($K_d = 33.7$ μ M by SPR and 5.78 μ M by ITC) could have such potent activity in biological assays (e.g. osteoclast inhibition $IC_{50} = 0.19$ μ M). Typically such a discrepancy between a binding affinity to its proposed therapeutic target and its biological activity would suggest that the real cellular target is not what was proposed. A discussion is needed to clarify this point.

Response: The comments are highly appreciated. Indeed, the K_d of ligand-receptor binding should be approximately proportional to the biological activity in cell-based inhibitory process. However, in our case, the simultaneously binding of sRANKL trimer with three RANK receptors are necessary for the formation of a functional sRANKL-RANK signal transduction complex to induce the osteoclastogenesis. It means that binding of a molecule of **S3-15** to anyone of the three surface clefts of sRANKL actually functionally blocks all the three sRANKL-RANK binding sites on sRANKL trimer, thus **S3-15**-sRANKL interaction cause a greater effect on

sRANKL-RANK interactions as well as a greater biological consequence.

In addition, such as regulating nucleic receptors, HSP90 proteins, or protein-protein interactions, the binding affinities are not necessarily proportionally to the biological activities. It is also observed that moderate binding affinities can amplify great biological activities (*J. Med. Chem.* 2021, 64, 2010–2023). RANKL-RANK interaction is the start-point of osteoclastogenesis. The interrupted interaction will result in the blockage of the entire osteoclastogenesis signaling pathways, such as NF- κ B, NFATC pathways (*J. Immunol.*, 2010, 184: 6910-6919). Therefore, the inhibition of **S3-15** on RANKL-RANK interaction can be amplified to a greater biological consequence.

These elucidations have been documented in the page 8 at line 11 of the manuscript. We copy the paragraph here for your review:

“**S3-15** binds to sRANKL with a modest binding affinity, however, the potency against osteoclastogenesis is high. In this case, the simultaneously binding of sRANKL trimer with three RANK receptors are necessary for the formation of a functional sRANKL-RANK signal transduction complex to induce the osteoclastogenesis. It means that binding of a molecule of **S3-15** to anyone of the three surface clefts of sRANKL actually functionally blocks all the three sRANKL-RANK binding sites on sRANKL trimer, thus **S3-15**-sRANKL interaction cause a greater effect on sRANKL-RANK interactions as well as a greater biological consequence.”

2.The authors have used cells and models from different species (rat and human) in different experiments. A clarification is needed on the sequence homology between human sRANKL and rat sRANKL.

Response: To clarify the sequence homology between human sRANKL and rat sRANKL, we have aligned the sequences of human sRANKL and rat sRANKL with BLAST. The result demonstrates that the identity and similarity between the two sRANKL were 89% and 93%, respectively. (see below)

Score	Expect	Method	Identities	Positives	Gaps
301 bits(772)	7e-112	Compositional matrix adjust.	142/159(89%)	149/159(93%)	0/159(0%)
Query 2	KPEAQFFAHLTINAANIPSGSHKVSLSWYHDRGWAKISNMTLSNGKLRVNQDGFYYLYA				61
Sbjct 2	K EAQFFAHLTINA +IPSGSHKVSLSWYHDRGWAKISNMT SNGKL VNQDGFYYLYA				61
Query 62	NICFRHHETSGSVPADYLQLMVYVVKTSIKIPSSHNLKGGSTKNWSGNSEFHFYSINVG				121
Sbjct 62	NICFRHHETSG + +YLQLMVYV KTSIKIPSSH LMKGGSTK WSGNSEFHFYSINVG				121
Query 122	GFFKLRAGEEISVQVSNPSLLDPDQDATYFGAFKVDID		160		
Sbjct 122	GFFKLR+GEEIS++VSNPSLLDPDQDATYFGAFKVDID		160		

We also compare the sequences of RANK binding area between human, mice and rat to see if this region is conserved. One research group has reported that the key residues in the RANKL - RANK binding region are conserved indeed (PANS, 2010, 107: 20281-20286) as shown in the following Figure. In total 20 residues, 18 of them are the same among three species.

3. From its chemical structure, S3, S3-15 and other analogues appear to be not stable. It is recommended that the authors provide stability data for these compounds in those key assays used in this study.

Response: The recommendation is appreciated. Actually, we did measure the stability of S3 and S3-15 with pharmacokinetics assay. As shown in Figure 2D, compound S3 and S3-15 represent a long T1/2 as 7.51 and 15.55 hours respectively.

Moreover, we also evaluated the stability of S3-15 structure as suggested. Two samples were tested, one is newly synthesized and another one is stored at 4°C without light for over 1 year. The HPLC experiment indicated that S3-15 is stable.

Here is the figures.

2. For modeling, the protonation state of those inhibitors need to be specified. For example, pyridine would be positively charged and the acid would be negatively charged.

Response: The comments are appreciated. Yes, the protonation states have been specified as shown in Figure 4 below.

Reviewer #3 (Remarks to the Author):

Manuscript NCOMMS-21-31271-T

Identifying Small Molecular Binding Site to Selectively Inhibit sRANK-RANK Interactions for Anti-osteoporosis Drug Discovery

A new compound, which binds to soluble RANKL (sRANKL) without binding to membrane bound RANKL (mRANKL) is created and tested in an ovariectomized (OVX) rats and mice model of osteoporosis.

The reviewer is highly appreciated for the positive remarks. The specific questions are answered as follows.

1. Page numbering is missing from the main document, but is present in the supplemental material.

Response: The page number has been added.

2. The Introduction needs to more carefully explain the need for selectively binding to sRANKL. Especially when it is taken into consideration that it has been shown that activated T cells both express mRANKL and secrete sRANKL (Immunology Letters 94, 239–246, 2004).

Response: The suggestion is accepted. The explanation needed for selectively binding to sRANKL has been documented in the page 2 at line 35 of the manuscript. Here is the added paragraph:

“In antigen presenting cells and T cells, mRANKL works as the receptor of RANK-RANKL reverse signaling pathway^{11, 12} and performs immune enhance functions, like T cell proliferation, T cell–dendritic cell interactions, DCs survival, thymus and lymph node development²³. Blocking mRANKL-RANK reverse interaction exhibits osteopetrosis as a result of a lack of osteoclast; defective T cell and B cell differentiation; and a failure of mammary gland lobuloalveolar development during pregnancy; decrease monocytes and DCs survival and their effective function²⁴⁻²⁶.”

Page 15, first paragraph

Bone Mineral Density (BMD) is not shown in any of the figures. Please replace with BV/TV (which is highly correlated to vBMD). Moreover, OVX + S3 had not

significantly higher BV/TV than OVX (Figure S7B, this relationship is marked “ns”).

Response: The suggestion is accepted. OVX + S3 actually is significantly higher BV/TV than OVX, and the Figure S7B is incorrectly labeled in the first submission. The “ns” is corrected to “*” in the revised manuscript.

Page 16, Figure 7

The 3D reconstructions are too small to see anything. You can easily omit BS/BV and TBPf to make more room in the figure. “VOX” -> “OVX”

Response: The suggestion is accepted. The figures of BS/BV and TBPf are removed. The 3D reconstructions pictures are enlarged. The typo is corrected.

Here is the revised Figures.

In Figure 7C, please start the y-axis in 0 (e.g. Tb.Th).

Response: The start of y-axis is changed to 0 in Figure 7C.

Please compare the ability of the identified compounds to counteract OVX-induced bone loss with that of denosumab at comparable doses (similar to Figure 6).

Response: The sequences of mice RANKL and human RANKL are different, hence, denosumab can only bind to human RANKL (J Bone Miner Res 2009; 24: 182 – 195; Biol. Pharm. Bull. 2018; 41: 637 – 643). Thus, the activity of denosumab cannot be measured in OVX-mice or OVX-rat. The efficacies of denosumab can only be

measured in monkey or human. Therefore, the abilities of the identified compounds and denosumab to counteract OVX-induced bone loss cannot be compared.

The influence on the trabecular microstructure is interesting, but the most important property is the bone fracture strength. Please report the mechanical strength of the bones of the rats and mice.

Response: Since the bone from previous study cannot be used for biomechanical test. In order to answer the reviewer's question, we have conducted a new *in vivo* experiment with mice. The mechanical strength of the femur of mice has been evaluated by three-point bending test in Figure S7F. The results demonstrated that **S3-15** treated mice exhibited significant improvement on ultimate load and strength compare to non-treatment (OVX). This result suggests that **S3-15** can increase the bone quality.

This experimental results have been documented in the page 17 at line 13 and Figure S7F of the manuscript. The added sentences are as follows:

“To further evaluate the biomechanical strength of the bones, a three-point bending test was conducted. The results indicated that the ultimate load and ultimate strength are correlated with the BV/TV after **S3-15** treatment (Figure S7F).”

Here is the Figure S7F.

Page 19, supplemental material

The 3D reconstructions are too small and BS/BV and TBPf can be omitted. If you want an expression of the curvature of the trabeculae, Structure Model Index (SMI) is

a better option than TBPf anyway.

Response: The suggestion is accepted. The figures of BS/BV and TBPf are removed as suggested. The 3D reconstructions pictures are enlarged. The CT instrument (Inveon PET/CT, Siemens, Germany) that used in this manuscript cannot provide SMI parameters.

In Figure S7 B + C “OVX +” is missing from the x-axis title.

Response: The “OVX+” has been added.

Page 35, micro-CT analyses

What do you mean with “After BMD detected”? A voxel size of 50 μm is much too small when the rat trabecular thickness is approximately 60 μm (Figure 7A) and the mouse trabecular thickness is approximately 40 μm for the OVX animals. According to the Nyquist sampling theorem the smallest object that can be resolved using a voxels size of 50 μm is 100 μm . This low spatial resolution may explain the look of the 3D reconstructions.

Response: We have removed this sentence from the “supporting material”. The typo has been corrected, the resolution is 19 μm .

Please, state the length of the integration time used and the threshold used for image segmentation (please follow the current guidelines when reporting micro-CT findings: Journal of Bone and Mineral Research 25(7):1468–1486, 2010).

Response: The suggestion is accepted. Based on the guidelines, we have corrected the procedure as follows:

“We used micro-CT to scan the femur from the femoral head to the femoral condyle, using 19 μm resolution, 80kV 500uA, 360 projection, 3000-6000 image threshold, full rotation cone beam” .

We have added this correction in the “Supplemental Data” in the page 37.

REVIEWERS' COMMENTS

Reviewer #1 (Remarks to the Author):

The authors have addressed my questions mostly and the paper has been improved. Regarding the use of the rat RANKL sequence, it will be informative to refer the original work on the rat RANKL sequence in this paper (PMID: 11092398).

Reviewer #2 (Remarks to the Author):

The authors have done an excellent job to address all the points raised by this reviewer. To address the issue that "However, it is surprising to see that S3-15 binds with sRANKL with a modest binding affinity ($K_d = 33.7 \mu\text{M}$ by SPR and $5.78 \mu\text{M}$ by ITC) could have such potent activity in biological assays (e.g.

osteoclast inhibition $\text{IC}_{50} = 0.19 \mu\text{M}$).", the authors responded by the following "However, in our case, the simultaneously binding of sRANKL trimer with three RANK receptors are necessary for the formation of a functional sRANKL-RANK signal transduction complex to induce the osteoclastogenesis. It means that binding of a molecule of S3-15 to anyone of the three surface clefts of

sRANKL actually functionally blocks all the three sRANKL-RANK binding sites on sRANKL trimer, thus S3-15-sRANKL interaction cause a greater effect on sRANKL-RANK interactions as well as a greater biological consequence."

Although this explanation is reasonable, it is still possible or even likely that S3-15 exhibits the anti-osteoporosis activity through additional mechanisms, in addition to its binding sRANKL and blocking the interactions of sRANKL-RANK. Please include this point in the Discussions.

Reviewer #3 (Remarks to the Author):

Concerning your mechanical data: In my view these important data should not be "hidden" in the supplementary data. Please move these data to the main document.

The F_{max} data are more or less as expected, although it is not uncommon that three point bending strength of mid-diaphyseal bone of the OVX'ed animals are equal to or greater than the sham operated animals. OVX of a rodent leads to a periosteal apposition and endocortical erosion (and thereby increased moment of inertia) leading to unaltered or even increased maximum force values. However, the "strength" data, which I gather is the maximum stress values are to me more perplexing. The maximum force values denotes the extrinsic bone strength, while the maximum stress values denotes the intrinsic bone strength i.e. the bone strength were the geometry has been taken into consideration. Please note that in mechanical compression the conversion from extrinsic bone strength to intrinsic bone strength is a matter of division with the cross sectional area. However, this conversion is much more complicated for three point bending (Bone 14(4): 595-608, 1993). To me it is puzzling that the differences between the groups looks so similar for the extrinsic and intrinsic bone strength. You do not need to rapport both extrinsic and intrinsic bone strength, I would suggest to only show the extrinsic bone strength (F_{max}).

You state that: "The results indicated that the ultimate load and ultimate strength are correlated with the BV/TV after S3-15 treatment". If you have made a correlation analysis please provide the correlation coefficients. Maybe you mean: "The mechanical strength of the femoral (or tibial ?) mid-diaphysis (Figure ??) provided similar results as the BV/TV of the distal femoral metaphysis (Figure ??)".